# Protein structural superfamily classification using hand-crafted and language model features: A performance *vs* interpretability trade-off

## Abstract

The newfound rise of protein language models (PLMs) that leverage data and compute has introduced an interesting conflict: a trade-off between the high predictive performance of non-interpretable features and the scientific insight that can be gained from interpretable, hand-crafted ones. In this work, we highlight and study this conflict via the task of classifying protein domains into their CATH superfamilies. We train one-vs-all (OvA) linear SVM classifiers for 45 *diverse* CATH superfamilies, each characterised by significant class imbalance. Our analysis compares *nine* feature vector types, which are either non-interpretable embeddings from PLMs or interpretable hand-crafted features. Our results demonstrate that PLM-based features achieve superior classification test scores of 90-99% with low variability, outperforming hand-crafted features by 20-30%. While PLM features yield high classification accuracy, their lack of interpretability obscures the underlying biological determinants. On the other hand, our novel structure-based *Contact Separation Interval Composition (CSIC) feature strikes an optimal balance. CSIC achieves highly competitive performance (~ 88%) with low overfitting while providing rich structural information about contact sequence separation.* Furthermore, we illustrate for two superfamilies, using the CSIC features and Marginal Contribution feature Importance (MCI) scores, that we can recover known structural characteristics of superfamilies such as characteristic long-range contacts and repeating amino acid motifs. This validates its utility for downstream applications, such as investigating protein-related diseases and guiding rational protein design.

## 1 Introduction

Proteins are often segmented into domains, which are subunits of a protein structure that fold independently of the rest of the structure (Kolodny et al., 2013). Studies estimate that the number of folds adopted by proteins in nature is between 1,000 and 10,000 (Kolodny et al., 2013). The CATH database categorises protein domains identified from PDB (Protein Data Bank) into hierarchical groups based on the similarity of their 3-dimensional fold. The protein domains in CATH are classified into 6,631 homologous superfamilies. We find many examples of sequences belonging to the same superfamily but having less than 35% sequence identity. The main questions that we seek to answer in this work are:

> *Can we predict the CATH superfamily of protein domains in an interpretable manner? Can we gain insights into the characteristic features that distinguish a given superfamily from others?*

We specifically target CATH superfamily classification as the superfamilies are defined by homologous protein domains that share significant structural similarity despite low sequence similarity (Orengo et al., 1997). In contrast, we do not consider function-based prediction tasks, like enzyme commission (EC) or gene ontology (GO) label prediction. This is because these classes often lack shared sequence or structural features (Omelchenko et al., 2010; Riziotis et al., 2025). Thus, the shared structural similarity in CATH superfamilies motivates our objective to find interpretable features that distinguish CATH superfamilies.

In this work, we compute nine different types of feature vectors from the sequence/structure of protein domains, and evaluate how well each type of feature vector can distinguish a given CATH superfamily from all others. We train linear classifiers to predict the CATH homologous superfamily of a protein domain using each of the nine different types of feature vectors. We do a robust study on 45 superfamilies curated based on the number of available sequences. This dataset of 45 superfamilies is diverse across various aspects (discussed in Section 3.1). A one-vs-all (OvA) linear support vector machine (SVM) classifier (with loss function capable of handling class imbalance) is trained using each type of feature vector to predict the CATH homologous superfamily of a protein domain. Although OvA classifiers are trained for only select 45 superfamilies, here each one-vs-all classifier implies 1-vs 6630 (superfamilies). The different feature vectors we use for training the classifiers can be categorized in two ways,

- sequence-based *vs* structure-based, and

- hand-crafted (interpretable) *vs* protein-language-model (PLM) based (non-interpretable)

*The nine feature vector types capture information at varying levels of granularity (coarse-grained to fine-grained) from the protein's sequence/structure.* The motivation here is to identify which type of feature vector, and thereby which level of information, is effective in distinguishing CATH superfamilies. We then use the best interpretable feature vector type in a downstream task to identify features that are characteristic of a CATH superfamily using a feature importance measure (Section 5.2). This can be useful in designing new proteins that are required to have a structure as characterised by a CATH superfamily.

Feature representation is an important aspect that contributes to the success of machine-learning methods, which are primarily data-driven. One attempts to translate the domain knowledge of a given learning task by defining features that are relevant and contribute to the learning task at hand. Also, this depends on the nature of the available dataset. Use of protein language models (PLMs) trained on large unlabeled datasets has become commonplace in computational biology (Pokharel et al., 2025; Weissenow & Rost, 2025). The PLM-based representations are high-dimensional and achieve high predictive performance on a wide range of tasks, which can be further improved with minimal fine-tuning (Weissenow & Rost, 2025). However, the uninterpretable nature of PLM-based representations and the inherent complexity of PLMs pose barriers to obtaining intelligible, actionable insights into the relationship between the input (protein sequence) and the output (prediction), thereby hindering the extension of domain knowledge. Many works highlight correlations between attention values and known protein properties (Simon & Zou, 2025; Vig et al., 2021). However, this is an emergent phenomenon from the self-supervised learning of the data manifold of available sequences rather than established causal relationships (see *'Limitations'* in Simon & Zou (2025)). As highlighted by recent debates in the field (Jain & Wallace, 2019; Pruthi et al., 2020; Hassid et al., 2022; Bibal et al., 2022), attention is not always explanation, and PLM embeddings lack direct, domain-knowledge-based interpretability by design. In this work, we invest in hand-crafted feature engineering from protein datasets and explore how well these interpretable features fare against uninterpretable PLM-based features in predictive performance on the CATH superfamily classification.

The main contributions of this work are as follows,

- *Trade-off analysis*: We highlight a trade-off between the predictive performance and interpretability of input features, using PLM-based features and hand-crafted features. The PLM-based features have high predictive performance but low/no interpretability, while hand-crafted features have relatively lower performance but high interpretability.

- *Novel structure features*: We propose two novel structure-based feature engineerings: OCPC (ordered contact pairs composition) and CSIC (contact separation interval composition). The features and dimensions of CSIC are determined by the distribution of the contact sequence separation of the superfamily, for which the OvA classification is performed.

- *Novel sequence features*: We propose a novel sequence feature engineering $k$OAAC ($k$-ordered amino acid composition), that encodes increasing levels of sequence order information with higher values of $k$

- *Novel PLM-based feature*: We propose a new feature engineering from the attention matrix of PLM: ProtBERT-Attn. This aggregates attention values by amino acid type.

- *Robust classification under imbalance*: Despite significant class-imbalance in OvA classification of superfamilies, we see high predictive performance for structure-based feature CSIC, comparable with PLM-based ProtBERT-Attn, while being significantly more interpretable.

- *Applicability to predicted structures*: We illustrate that CSIC features computed from both experimentally determined and predicted structures (AlphaFold, Jumper et al. (2021)) have comparable predictive performance in the OvA classification of superfamilies.

- *Recover established superfamily characteristics*: We present two case studies where we recover known characteristic features of a superfamily from interpretable hand-crafted CSIC and AAC features. In these studies, we infer characteristic features of superfamilies, such as long-range contacts, amino acids present in repeating motifs, and contacts corresponding to anti-parallel $\beta$-strands. For this, we rely on the marginal contribution feature importance (MCI) score Catav et al. (2021).

We discuss our feature engineering in detail in Section 2. Details of the dataset used are in Section 3. The methodology for training/evaluation of classifiers is in Section 4. We discuss the results of our computational experiments in Section 5, and our conclusions in Section 6. Many details are in Appendices A.1-A.9.

## 2  Feature Engineering

We use broadly three types of feature vectors engineered from the protein domain, *which encode different levels of information.* For computing these features, we use the standard 20 amino acid types, (A, R, N, D, C, E, Q, G, H, I, L, K, M, F, P, S, T, W, Y, V), we refer to these using $\mathcal{T} = \{t_1, t_2, \cdots, t_{20}\}$. We briefly describe each of the feature engineerings below. Figure 1 and Table 1 provides a summary of these features. More details, including mathematical definitions, are in Section A.2.

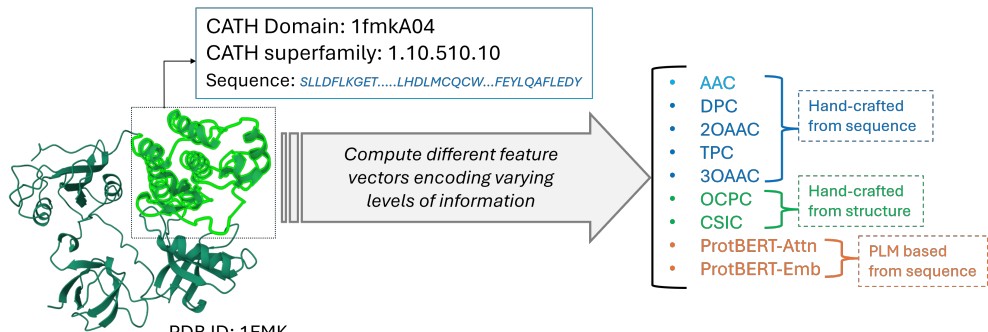

Figure 1: Nine types of feature vectors computed from a protein domain. See Table 1 for details.

### 2.1  Hand-crafted features from sequence

From the sequence, we compute one type of feature vector that doesn't utilise any sequence order information and four other types of feature vectors that encode varying levels of sequence order information. These are discussed below.

**Amino acid composition (AAC).** As a simplistic feature, we count the occurrences of each of the 20 amino acid types. This results in a 20-dimensional feature vector.

The AAC feature completely ignores the amino acids' order in the sequence. Two protein sequences $\mathbf{p}$ and $\mathbf{q}$ will have the same amino acid composition if $\mathbf{q}$ is a permutation of $\mathbf{p}$. Thus, we introduce the $k$-ordered amino acid composition ($k$OAAC) feature vector, which considers the amino acids' relative order in the sequence. We discuss this in detail below.

Table 1: A summary of the feature vector types computed from a protein domain's sequence/structure.

| Feature Engineering | Dimension | Feature vector component description/interpretation |
|---|---|---|
| *Hand-crafted, from sequence* | | |
| Amino acid composition (AAC) | 20 | Number of times an amino acid type $t_i$ occurs in the sequence. Each dimension corresponds to a different amino acid type. |
| Dipeptide composition (DPC) | $20^2 = 400$ | Number of times amino acid type pairs $(t_{i_1}, t_{i_2})$ occur adjacent to each other in the sequence in that order. Each dimension corresponds to a different ordered-pair of amino acid types. |
| Tripeptide composition (TPC) | $20^3 = 8000$ | Number of times amino acid types $(t_{i_1}, t_{i_2}, t_{i_3})$ occur adjacent to each other in the sequence in that respective order. Each dimension corresponds to a different ordered triplet of amino acid types. |
| 2-ordered amino acid composition (2OAAC) | $20^2 = 400$ | Out of the $\binom{L}{2}$ ordered position pairs $(p_{j_1}, p_{j_2}), j_1 < j_2$, in the sequence, the number of such position pairs having amino acid types $(t_{i_1}, t_{i_2})$ in that respective order. Each dimension corresponds to a different ordered pair of amino acid types. |
| 3-ordered amino acid composition (3OAAC) | $20^3 = 8000$ | Out of the $\binom{L}{3}$ ordered position triplets $(p_{j_1}, p_{j_2}, p_{j_3}), j_1 < j_2 < j_3$, in the sequence, the number of position triplets having amino acid types $(t_{i_1}, t_{i_2}, t_{i_3})$ in that respective order. Each dimension corresponds to a different ordered-triplet of amino acid types. |
| *Hand-crafted, from structure* | | |
| Ordered contact pairs composition (OCPC) | $20^2 = 400$ | Number of times the amino acid type pair $(t_{i_1}, t_{i_2})$ are in contact in the structure and occurs in the sequence in the same relative order. Each dimension corresponds to a different ordered pair of amino acid types. |
| Contact separation interval composition (CSIC) | $K \times 20$ ($K$ is determined from the data) | Number of contacts an amino acid type $t_i$ has in the structure with another amino acid separated by at least $l$ and at most $u$ residues in the sequence. Each dimension corresponds to a different amino acid type $t_i$ and interval $(l, u)$ combination. $K$ is the number of such intervals considered. |
| *Protein language model (PLM) based, from sequence* | | |
| ProtBERT-Emb | 1024 | Averaged embeddings of the final layer of protein language model Prot-BERT. No interpretation for dimensions. |
| ProtBERT-Attn | $16 \times 20 = 320$ | Each dimension is the aggregation of the row-sum of the attention-matrix for the rows corresponding to amino-type $t_1$. This is done for each attention-head (total 16). The attention matrix is from the final layer of ProtBERT. No interpretation for attention-values. |

**Features that encode sequence order.** We use 4 types of features that encode sequence order information, partially, into the feature vector dimensions by accounting for the relative order/position of amino acids in the protein sequence. These are *dipeptide composition (DPC), tripeptide composition (TPC), 2ordered amino acid composition (2OAAC)* and *3ordered amino acid composition (3OAAC)*. DPC and TPC are existing and widely used features, while 2OAAC and 3OAAC are novel feature engineerings that are introduced in this work.

DPC is a $(20^2 =)$ 400-dimensional feature that computes the count of the contiguous 2-mers of given amino acid types in the sequence. Similarly, TPC is a $(20^3 =)$ 8000-dimensional feature that computes the count of contiguous 3-mers of given amino acid types in the sequence.

We introduce two novel features that encode sequence order information, 2OAAC and 3OAAC. 2OAAC is similar to DPC but allows any number of residues (can be even 0) between the two amino acids, with the order of the two amino acids maintained. Similarly, the $20^2 = 400$ dimensional 2OAAC feature vector can be computed by counting the occurrence of all $20^2$ ordered pairs of amino acids. Likewise, 3OAAC is similar to TPC but allows any number of residues (including 0) between the three amino acids, with the order of the three amino acids maintained. In general, for $k$OAAC, the feature dimension $i$ corresponding to the ordered

tuple $\left(t_{i_1}, t_{i_2}, \cdots, t_{i_k}\right)$ of amino acid types, for a sequence $\mathbf{p}$ can be computed as,

$$
\begin{aligned}
x_i^{kOAAC} &= x_{(i_1, i_2, \cdots, i_k)}^{kOAAC}, \qquad i = i_1 + \sum_{r=2}^{k} 20^r (i_r - 1) \in \left[20^k\right] \\
&= \sum_{1 \le j_1 < j_2 < \cdots < j_k \le L} \mathbf{1}_{\{\mathrm{p}_j = \mathrm{t}_{i_1}, \mathrm{p}_{j+1} = \mathrm{t}_{i_2}, \Pi, \mathrm{p}_{j+k-1} = \mathrm{t}_{i_k}\}}
\end{aligned} \tag{1}
$$

## 2.2 Hand-crafted features from structure

We propose two types of *novel feature vectors* from the 3D structure of the protein domains. Availability of high-accuracy predicted 3D structures of proteins makes it possible to compute these vectors. In particular, Alphafold has provided high-accuracy 3D structures for most proteins, which makes it possible to compute feature vectors that we are proposing here. For computing these features, we first compute a contact map ($C$) from the protein's structure. Please see Section A.2.2 for details. We use the contact map of the protein domain to compute the two types of structure-based feature vectors that are discussed below.

**Ordered contact pairs composition (OCPC).** We define OCPC as a $(20^2 =)$ 400-dimensional feature that computes the count of contacts formed by given pairs of amino acid types in the protein structure. Here, the contacts are defined by the contact map. The relative order in which the two amino acids defining the contact occur in the sequence is also considered. The OCPC feature dimension $i$ for the amino acid type pair $\left(t_{i_1}, t_{i_2}\right)$ from protein $\mathbf{p}$ with its contact map $C$ is computed as follows,

$$
x_i^{OCPC} = \sum_{1 \le j_1 < j_2 \le L} \mathbf{1}_{\{\mathrm{p}_{j_1} = \mathrm{t}_{i_1}, \mathrm{p}_{j_2} = \mathrm{t}_{i_2}\}} \times C_{j_1, j_2}, \qquad i = i_1 + 20(i_2 - 1) \in \left[20^2\right] \tag{2}
$$

Thus, the feature dimensions of OCPC contain two kinds of information. One is the amino acid type pairs that are in contact in the 3-dimensional structure of the protein, and the other is the relative order in which these contact-forming amino acid pairs occur in the sequence.

**Contact separation interval composition (CSIC).** We define CSIC as $K \times 20$ dimensional feature that counts the number of contacts a given amino acid type has with any other amino acid that is within a given sequence separation range. The sequence separation intervals/ranges can be a user-defined set, $\mathcal{I} = \{[l_1, u_1], [l_2, u_2], \cdots, [l_K, u_K]\}$. Here, $K$ is the size of $\mathcal{I}$ as defined by the user. We define this set $\mathcal{I}$ in a data-driven manner (discussed in Section 4.1.2). Let the $K$ intervals defined by the user be, $\mathcal{I} = \{[l_1, u_1], [l_2, u_2], \cdots, [l_K, u_K]\}$. The CSIC feature dimension $i$ for the amino acid type $t_{i_1}$ and interval $[l_k, u_k]$ from protein $\mathbf{p}$ with its contact map $C$ is computed as follows,

$$
\begin{aligned}
x_i^{CSIC} &= x_{i_1, (l_k, u_k)}^{CSIC}, \qquad i = i_1 + 20(k - 1) \in \left[K \times 20\right] \\
&= \sum_{1 \le j_1 < j_2 \le L} C_{j_1, j_2} \times \mathbf{1}_{\{l_k \le j_2 - j_1 \le u_k\}} \times \mathbf{1}_{\{p_{j_1} = t_{i_1} \vee p_{j_2} = t_{i_1}\}}
\end{aligned} \tag{3}
$$

As in OCPC, the feature dimensions of CSIC contain two kinds of information. One is the number of contacts an amino acid type forms with other amino acids in the protein's 3-dimensional structure. The other is how separated in the sequence are these amino acids that form contacts.

## 2.3 Protein language model (PLM) based features from sequences

We compute two types of feature vectors using a pre-trained PLM, ProtBERT (Elnaggar et al., 2021).

Given an input protein sequence $\mathbf{p}$ of length $L$, ProtBERT returns $L$ number of 1024-dimensional embedding vectors corresponding to each position of the input sequence. This can be viewed as a $L \times 1024$ matrix. We take the average of this matrix along the sequence length dimension to get a single 1024-dimensional embedding vector for the input sequence $\mathbf{p}$. We refer to this feature vector type as *ProtBERT-Emb*. The feature dimensions of ProtBERT-Emb lack a domain-knowledge-based interpretable notion.

Another feature vector that we compute from ProtBERT is using the attention-matrix from its final layer. Each layer of ProtBERT has 16 attention-heads, each generating an attention-matrix. We compute a 320-dimensional ($16 \times 20$) feature matrix that aggregates attention values by amino acid type. We refer to this

feature vector type as *ProtBERT-Attn.* The ProtBERT-Attn feature dimension $i$ for amino acid type $t_{i_1}$ and attention-head $h$ is computed as follows,

$$x_i^{\text{ProtBERT-Attn}} = x_{(i_1,h)}^{\text{ProtBERT-Attn}}, \qquad i = i_1 + 20(h-1) \in [16 \times 20]$$

$$= \sum_{j_1=1}^{L} \left( \sum_{j_2=1}^{L} A_{j_1,j_2}^h \right) \times \mathbf{1}_{\{p_i = t_{i_1}\}} \tag{4}$$

Although the feature dimension for ProtBERT-Attn is defined by amino acid types, the attention values aggregated do not have a domain-knowledge-based interpretation.

Here, ProtBERT is used as a representative PLM to illustrate the high classification performance that can be achieved on this task. As shown in Table 2, using ProtBERT, an average test score of 96.5% is obtained. This is already a high score, and using any other PLM can achieve rather minor - 4.5% - improvement on this. We believe the performance of ProtBERT sufficiently demonstrates the high predictive power of PLMs, while the notion of interpretability is largely similar across popular PLMs. That is by finding correlations between attention values and known protein properties (Vig et al., 2021; Simon & Zou, 2025). So, other PLMs were not used in this study.

## 3 Datasets and their key characteristics

The CATH database (Sillitoe et al., 2020) categorises more than 0.5 million protein domains at four hierarchical classification levels: Class → Architecture → Topology → Homologous superfamily. Class is based on the predominant secondary structure component in the fold. Architecture is based on the relative arrangement of secondary structures in the 3-dimensional space. Topology is based on how the secondary structure components are connected in a fold. Homologous superfamily is based on evidence of common ancestry

The protein domains in the CATH database are classified into 6,631 homologous superfamilies. We use non-redundant datasets of homologous superfamilies with 35% as the sequence identity threshold (downloaded from the CATH website); in all, this includes 32,388 CATH domains. We train a binary classifier for each superfamily that has at least 100 representative domain sequences, i.e., for 45 superfamilies. Figure 2 shows the number of representative domains and the sequence-length distribution for each of the 45 superfamilies.

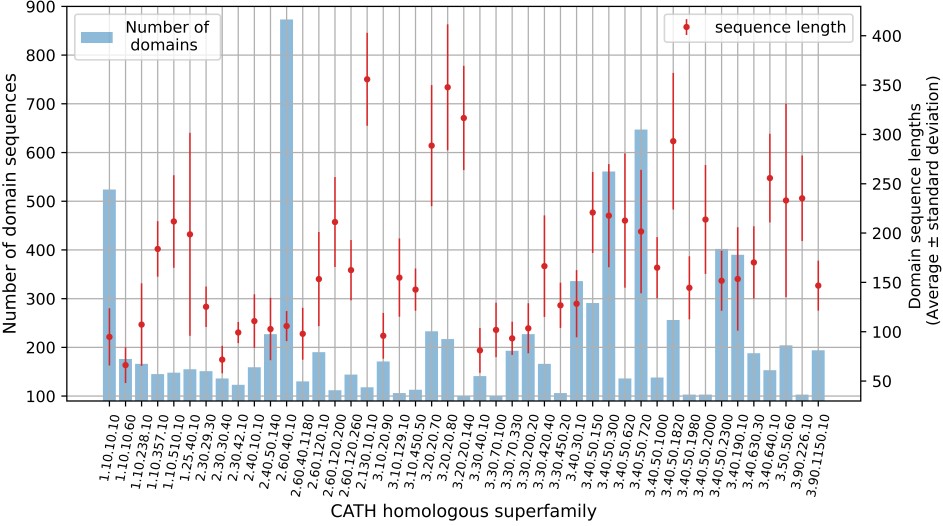

Figure 2: (*Illustrating dataset diversity*) The bar plot shows the number of representative domain sequences (left y-axis) available for the selected 45 CATH homologous superfamilies in the non-redundant dataset with a 35% sequence identity threshold. The scatter plot with error bars shows the distribution of the length of the domain sequences (right y-axis) for each superfamily.

### 3.1 Dataset diversity

The selected datasets for 45 superfamilies are diverse due to:

- *Sequence diversity*: No two sequences have more than 35% sequence identity

- *Structural diversity*: The 45 superfamilies span 'mainly alpha' (6) i.e. CATH ID 1.*, 'mainly beta' (11) i.e. CATH ID 2.*, and 'alpha beta' (28) i.e. CATH ID 3.*, classes in the 1st level of CATH hierarchical classification (Figure 2). Alpha and beta denote secondary structure patterns.

- The dataset size for a given superfamily varies from 100 to 873 sequences (Figure 2).

- There is no correlation between the variation of sequence lengths and the number of representative CATH domain sequences of a given superfamily.

More details in Section A.1.

## 4 Classifiers for CATH superfamily prediction and MCI feature importance computation

We train one-vs-all classifiers to predict the CATH homologous superfamily of a protein domain.

### 4.1 One-vs-all linear SVM classifiers

For each of the selected 45 superfamilies described in Section 3, we train a one-vs-all binary classifier predicting whether a given domain sequence belongs to the corresponding superfamily or any of the other 6,630 CATH homologous superfamilies. For each classifier, the positive class dataset comprises protein domain sequences from one of the 45 superfamilies, and the negative class dataset comprises domain sequences from all the other 6,630 superfamilies. The train and test set for a classifier is made with an 80:20 split of both positive and negative datasets.

#### 4.1.1 Training and evaluation with class-imbalance

If there are $m_1$ samples in the positive dataset, then the negative dataset has $m_2 = 32,388 - m_1$. We have $m_1 \in [100, 873]$ across the selected datasets (Figure 2), therefore the range of class-imbalance ratio is $m_1/m_2 \in [0.003, 0.028]$. Thus, each classifier's train/test data has a large class imbalance (an average imbalance of 1:197). To account for this, the test performance of a classifier was measured using the Arithmetic Mean (AM) of specificity and sensitivity (Brodersen et al., 2010). Also, an empirically class-balanced version of squared hinge loss is used in training the SVM as suggested in Menon et al. (2013) for statistical consistency with the AM score. For each classifier, 10% of the train set is used as a validation set for tuning the SVM regularisation hyperparameter $C$ (please see Section A.4 for details). $C$ is inversely proportional to the strength of regularisation. The average AM scores are reported with 5 random train/test splits for each superfamily and each feature.

Scikit-learn's (Pedregosa et al., 2011) `LinearSVC` module is used for training the classifiers for all features except TPC and 3OAAC. As TPC and 3OAAC features are very high-dimensional, we used scikit-learn's `SGDClassifier` module with hinge-loss and mini-batch size of 10,000 for training the linear classifiers using these features.

#### 4.1.2 CSIC intervals for one-vs-all CATH superfamily classification

Recall from Section 2.2 that for CSIC feature computation, a user-defined input, i.e. a set of sequence separation intervals $\mathcal{I}$ is required. We define $\mathcal{I}$ in a data-driven manner based on the superfamily for which the OvA classifier is trained. For doing say, superfamily '1.10.10.10'-vs-'other' classification, we first look at the empirical distribution of the sequence separation of the contact residues of all the structures of this family in the training split. See Figure 3. Contacts by residues that are adjacent in the sequence are ignored.

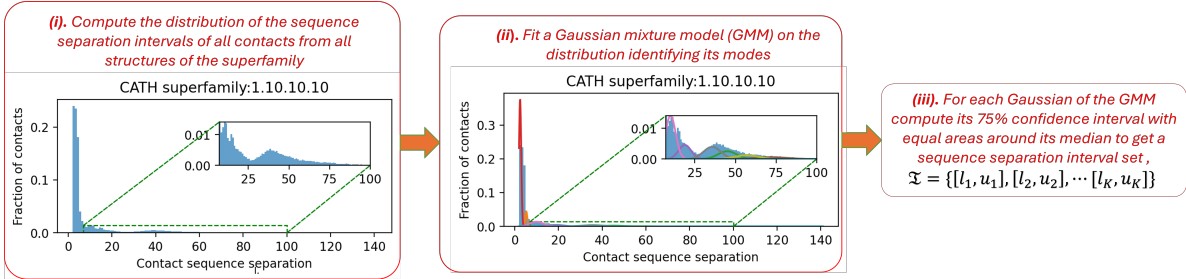

Figure 3: Steps for computing the set of sequence separation intervals $\mathcal{I}$ in a data-driven manner for CSIC-Gaussian feature computation.

We see the distribution is light-tailed with the highest concentration at 2. Zooming in on the tail, we see that the distribution has many small modes. These distinct modes need not be random noise and could correspond to conserved mid-range and long-range structural contacts that define the characteristic 3D fold of the superfamily. To infer the multiple prominent modes in the distribution, we approximate it using Gaussian mixtures (Bishop, 2006). The mean of each fitted Gaussian component identifies the expected sequence separation distance of a conserved structural motif. The standard deviation of each component naturally defines a statistical tolerance interval around the motif, accounting for evolutionary variations such as residue insertions or deletions. From each of the fitted Gaussians, we compute the 75% confidence interval with equal areas around the mode. Thus, if $K$ Gaussians were fitted, we get $K$ intervals which we use as $\mathcal{I}$. We refer to this feature, where CSIC intervals are computed using Gaussian mixtures, as CSIC-Gaussian. Since the contact separation distribution has a semi-infinite support, we also used gamma mixtures (Xiong et al., 2024) for defining $\mathcal{I}$. We refer to this feature as CSIC-Gamma. Please see Section A.2.4 for additional implementation details.

## 4.2  Feature importance via MCI

Marginal Contribution feature Importance (MCI) is an axiomatic feature importance score proposed to explain data (Catav et al., 2021). To compute MCI scores for features in a feature set $N$, we first define a value function $v(S)$ for every feature subset $S \subseteq N$. We define $v(S)$ as a measure of linear separation between the (binary) classes in the feature space of $S$ (this is adapted from Tripathi et al. (2020)). Accounting for class-imbalance, we define $v(S)$ using a class-balanced hinge loss function $tr\_er(S)$, which is defined as,

$$tr\_er(S) = \min_{w,\xi_j} \frac{1}{2n_+} \sum_{j=1}^{n_+} \xi_j + \frac{1}{2n_-} \sum_{j=n_++1}^{n_-} \xi_j \tag{5}$$

$$\text{s.t. } y_j \left( \sum_{i \in S} w_i x_{j,i} + b \right) \geq 1 - \xi_j, \ \forall j \in \left[ n_+ + n_- \right] \tag{6}$$

$$\xi_j \geq 0, \ \forall j \in \left[ n_+ + n_- \right] \tag{7}$$

and $v(S) = tr\_er(\varnothing) - tr\_er(S)$. $n_+$ and $n_-$ are the number of training samples in each class (binary). $\{(x_j, y_j)\}_{j=1}^{n_+ + n_-}$ is the training data, with $x_j \in \mathbb{R}^{|N|}$ and $y_j \in \{-1, +1\}$. The minimizer in the above finds a linear hyperplane with the least class-balanced hinge loss in the feature space of $S$. $\varnothing$ is the empty set and $tr\_er(\varnothing) = 1$, therefore, $v(S) = 1 - tr\_er(S)$. $tr\_er(S) = 0$ implies $v(S) = 1$, i.e., the two classes are completely linearly separable in the feature space of $S$. The maximum value of $tr\_er(S)$ possible is 1.

For a feature $i \in N$, its MCI score is defined as,

$$MCI(i) = \max_{S \subseteq N \setminus \{i\}} v(S \cup \{i\}) - v(S). \tag{8}$$

Exact feature-wise MCI computation requires evaluating the value function over $2^{|N|}$ subsets. Hence, they are computed using a linear time (in number of features) Monte Carlo sampling based approximation (Castro et al., 2009), similar to Shapley value approximation. However, the bounds on the approximation's variance degrade significantly as $|N|$ increases, leading to unstable feature rankings. Thus, for high-dimensional features like CSIC, instead of computing MCI for each feature, we partition the feature set $N$ and compute a collective MCI score for each partition. This strategy is discussed below.

### 4.2.1 Row-wise / column-wise MCI for CSIC features.

Recall that the CSIC features can be viewed as a $K \times 20$ matrix (see Table 1), where the $K$ rows correspond to sequence separation intervals and the 20 columns correspond to amino acid types. We naturally obtain interpretable row-wise and column-wise partitions of the matrix. If $N$ is the set of all $K \times 20$ features, we partition $N$ row-wise as $N_R = \{R_1, R_2, \cdots R_K\}$ to compute the row-wise MCI. Here, each $R_i$ is a set of 20 features corresponding to row $i$ of CSIC feature matrix. The MCI for row $i$ is then computed as follows,

$$MCI(i) = \max_{S_R \subseteq N_R \setminus \{R_i\}} v\left(\bigcup_{R_j \in S_R} R_j \cup R_i\right) - v\left(\bigcup_{R_j \in S_R} R_j\right) \qquad (9)$$

For column-wise MCI we partition $N$ column-wise as $N_C = \{C_1, C_2, \cdots, C_{20}\}$, where $C_k$ is a set of $K$ features corresponding to the column $k$ of CSIC feature matrix. The MCI for column $k$ is then computed as follows,

$$MCI(k) = \max_{S_C \subseteq N_C \setminus \{C_k\}} v\left(\bigcup_{C_j \in S_C} C_j \cup C_k\right) - v\left(\bigcup_{C_j \in S_C} C_j\right) \qquad (10)$$

Thus, for computing row-wise MCI, we only require evaluating the value function over $2^K$ subsets, which is much less than $2^{|N|} = 2^{K \times 20}$. Similarly, for column-wise MCI. The limitation of this approach is that we do not obtain a feature importance score for each individual feature, but rather a collective score for each subset of features defined by the partitions. All features within a partition need not have equal importance, and the relative importance between two individual features from different partitions is also not determined. Hence, the partition-based MCI scores are a coarser measure of feature importance than individual MCI scores.

## 5 Classification results and interpretability insights

### 5.1 Predictive performance

Table 2: The average train/test AM scores over 5 random splits are again averaged across the 45 superfamilies. Similarly, the standard deviations (s.d.) of train/test AM scores over 5 random splits are again averaged across the 45 superfamilies. The dimensions of each type of feature vector are given in parentheses. See Section A.7 for validation scores and accuracies.

| AM | | Hand-crafted sequence-based | | | | | | Hand-crafted structure-based | | PLM-based | |
|---|---|---|---|---|---|---|---|---|---|---|---|
| | Avg. | AAC (20) | DPC (400) | 2OAAC (400) | TPC (8000) | 3OAAC (8000) | OCPC (400) | CSIC-Gauss ($K\times20$) | CSIC-Gamm ($K\times20$) | PB-Attn (320) | PB-Emb (1024) |
| Train | Avg. | 81.7 | 96.1 | 92.5 | 93.6 | 83.7 | 94.7 | 97.2 | 97.0 | 97.1 | 99.7 |
| | (s.d.) | (0.9) | (1.8) | (3.1) | (5.0) | (5.4) | (1.8) | (1.5) | (1.5) | (1.9) | (0.2) |
| Test | Avg. | 79.8 | 75.0 | 79.0 | 77.8 | 77.4 | 79.2 | 88.3 | 87.8 | 88.5 | 96.5 |
| | (s.d.) | (2.4) | (6.1) | (4.7) | (5.3) | (4.5) | (4.6) | (4.3) | (4.4) | (3.8) | (2.0) |

Figure 4 and Table 2 report the one-vs-all classification scores for the 10 feature vectors across 45 superfamilies. Our main observations are:

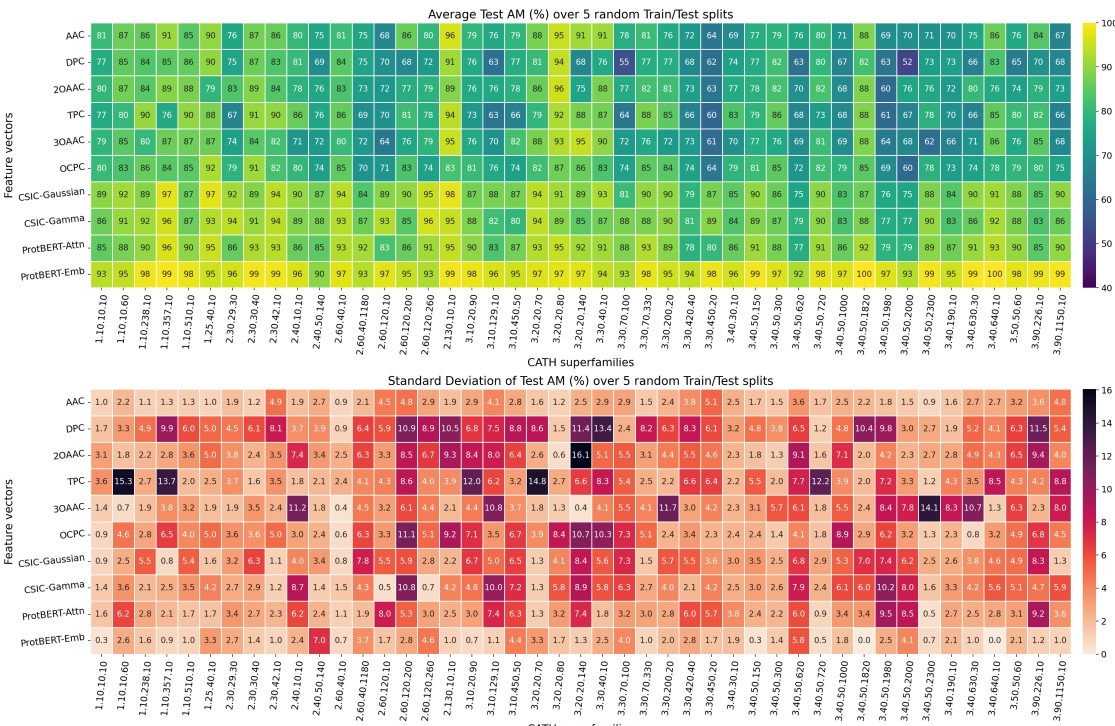

Figure 4: Heatmaps of average and standard deviations of test AM scores across 5 random train/test splits for the 45 superfamilies using each of the 10 feature vectors. See Section A.6 for train/val.

- Among the 10 different feature vectors considered, the PLM-based feature ProtBERT-Emb exhibits the highest predictive performance (>90%) across the 45 superfamilies. The standard deviation of test scores across random splits is also the least for ProtBERT-Emb.

- AAC feature, which does not use any sequence order information, has >60% test AM scores across the 45 superfamilies. The test AM is >80% for 20 superfamilies and >90% for 6 superfamilies. The standard deviation of test scores across random splits is also low.

- All hand-crafted features except AAC have high standard deviations of test scores across random splits. We find that this is particularly high for superfamilies with less number of samples, leading to greater class imbalance in OvA classification. Please see Figure 5. We demonstrate in Appendix A.3.4 that applying PCA-based dimensionality reduction effectively controls this variance and stabilizes the test scores across random splits.

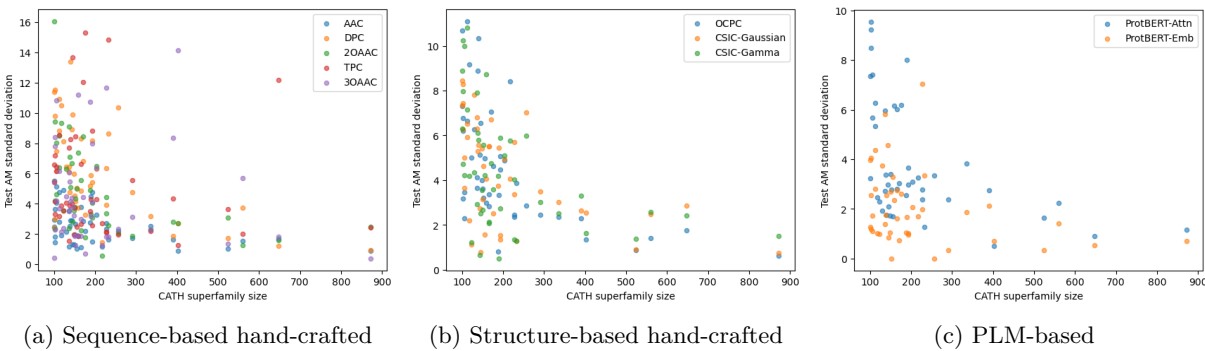

(a) Sequence-based hand-crafted     (b) Structure-based hand-crafted     (c) PLM-based

Figure 5: The standard deviation of test AM scores v/s the OvA classification superfamily size.

- The DPC, 2OAAC, TPC, and OCPC features have significant overfitting, as can be seen from the difference between train and test AM scores.

- Hand-crafted structure-based CSIC features, with low overfitting, have performance comparable with ProtBERT-Attn; the average test scores across superfamilies being $\approx 88\%$.

- We observed increased overfitting when combining DPC/2OAAC/TPC/3OAAC features. When combining AAC and CSIC features (the hand-crafted features with the least overfitting), we did not see a significant difference in test scores.

***On overfitting with hand-crafted features.*** Generalisation of classification performance on the test set is challenging as the train and test sequences have low identity (<35%). However, the low overfitting with ProtBERT features could be due to a good approximation of the naturally occurring sequence data manifold via pre-training on all available sequences (including our test sequences).

We validated the statistical significance of model performance and the applicability of CSIC features to predicted structures (details in Section A.3). Our main observations are:

- ***Statistical significance of performance differences***: We performed bootstrapping on the test set to compute the statistical significance of test AM score differences of the linear SVMs trained using different feature types. We observe that the ProtBERT-Emb feature consistently outperforms other features across the 45 superfamilies. More details are in Section A.3.2. ProtBERT-Attn and CSIC features perform comparably across the 45 superfamilies, while the rest of the hand-crafted features have relatively lower classification performances compared to them.

- ***Applicability of CSIC to predicted structures***: We created a dataset of 5350 AlphaFold predicted structures containing all 45 superfamilies and others. We tested the OvA classifiers on CSIC-Gaussian features computed from these structures. More details are in Section A.3.3. The average classification AM score across the 45 superfamilies is $85.2 \pm 6.46$ (mean $\pm$ standard deviation). Thus, we do not observe a significant drop in classification performance when the CSIC feature is computed from AlphaFold structures.

***Error analysis.*** To better understand the classification behavior, we analyzed the per-class confusion matrices for the OvA classifiers trained with CSIC-Gaussian features (detailed plots are provided in Appendix A.9). While the False Positive Rates (FPRs) remain low across the 45 superfamilies, the sheer volume of the negative class (averaging a 1:197 imbalance ratio) means that the absolute errors are heavily dominated by False Positives (FPs).

In our OvA setup, a sample from the negative 'other' class can belong to any of the remaining 6,630 CATH superfamilies. However, our analysis reveals that these FPs are not random misclassifications. Below are our observations from our analysis of FPs from one of the test splits:

- For 18 of the analyzed 45 superfamilies, the highest number of FPs originates from 'other' class domains that share the exact same *Topology* (the 'T' label in the CATH hierarchy, representing the same 3D structural fold). Given that there are 831 distinct topologies in the test set, the probability of the model assigning the majority of its FPs to the correct topology by random chance is extremely low ($< 0.0013$).

- We note that for 10 of the remaining 27 superfamilies, there are fewer than 5 'other' class samples available within their respective topologies, which heavily restricts the model from making this specific type of error.

- For 7 of the remaining 17 superfamilies, the highest number of FPs belongs to the same *Architecture* (the 'A' label, representing a similar spatial arrangement of secondary structures). With 38 distinct architectures present in the test set, a random assignment to the correct architecture is also highly improbable ($< 0.027$).

This error profile shows that CSIC-Gaussian feature-based linear SVM misclassifications are not random and often occur between structurally similar superfamilies. Furthermore, the low FPRs demonstrate that the aggregated structure information in the hand-crafted CSIC features are sufficiently nuanced to distinguish a given CATH superfamily from many others.

## 5.2 Recovering structural characteristics of a superfamily via feature interpretability

We derive biological insights, known structural characteristics, from the OvA classification using hand-crafted features for two superfamilies. For this, we use marginal contribution feature importance (MCI) scores (Catav et al., 2021) as discussed in Section 4.2. We used the entire train+test data for determining the CSIC-Gaussian intervals in the examples below.

### 5.2.1 Long-range contacts in superfamily 3.40.640.10

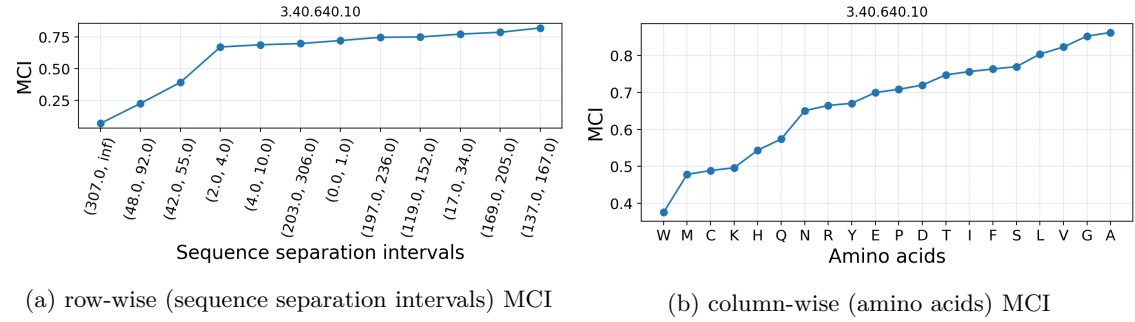

(a) row-wise (sequence separation intervals) MCI

(b) column-wise (amino acids) MCI

Figure 6: Row and column-wise MCI scores of CSIC-Gaussian $K \times 20$ feature matrix, for 3.40.640.10 vs 'others' classification.

Recall that CSIC features are $K \times 20$ dimensional, with rows representing sequence separation intervals and columns representing amino acid types (see Table 1). Since MCI approximations degrade in high dimensions, we compute row-wise and column-wise MCI of the $K \times 20$ feature matrix. See Figure 6. The intervals [137, 167] and [169, 205] have the highest row-wise feature importance scores. These are long-range contacts present in the structures of this superfamily as highlighted in the contact maps, see Figure 7. A study (Deu et al., 2002) highlights the role of a long-range contact that falls within this range [137, 205], in the structure of an aspartate aminotransferase domain that belongs to 3.40.640.10.

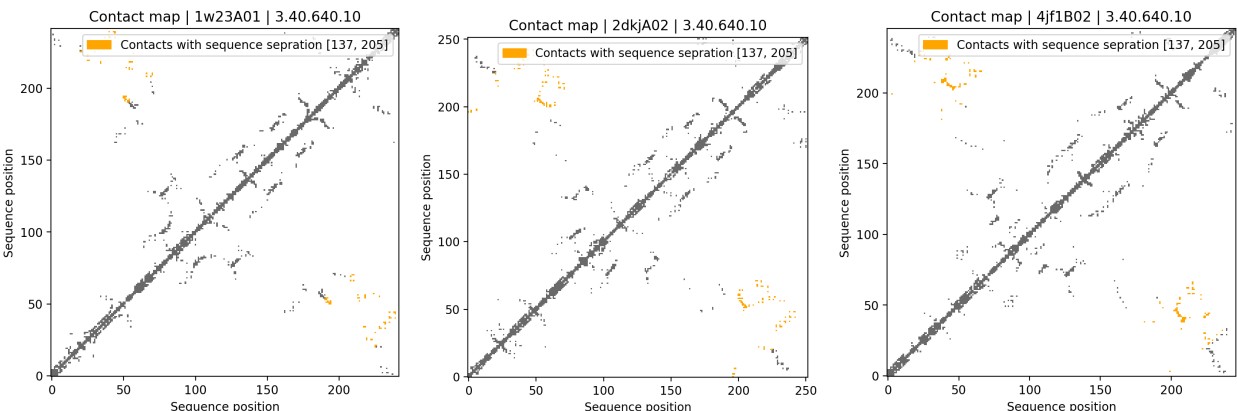

Figure 7: Contact map for 3 protein domain structures belonging to CATH superfamily 3.40.640.10. More are available in Section A.5 Figure 18.

### 5.2.2 Characteristic short-range contacts and amino acids from repeating motifs in superfamily 2.130.10.10

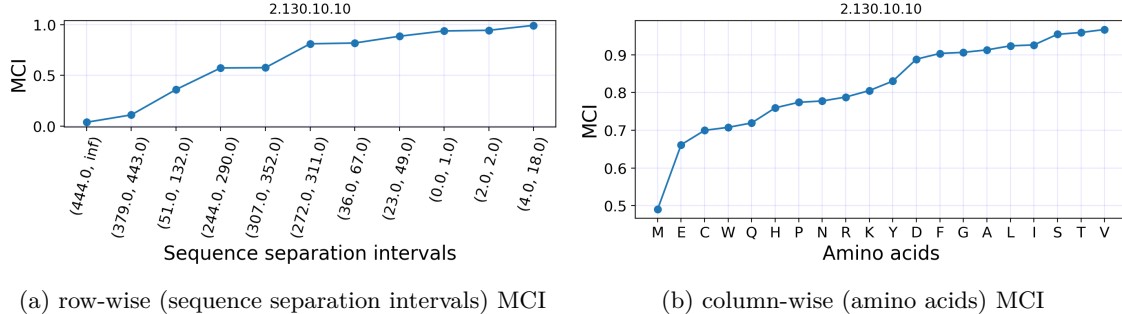

(a) row-wise (sequence separation intervals) MCI      (b) column-wise (amino acids) MCI

Figure 8: Row and column-wise MCI scores of CSIC-Gaussian $K \times 20$ feature matrix, for 2.130.10.10 vs 'others' classification.

**Using CSIC-Gaussian.** As in the previous example, we compute row-wise and column-wise MCI feature importance of the $K \times 20$ feature matrix. See Figure 8.

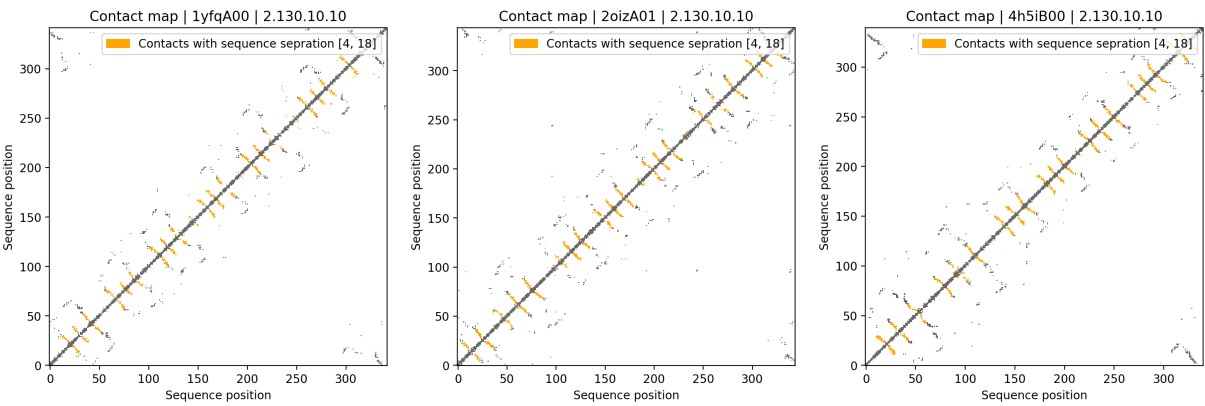

Figure 9: Contact map for 3 protein domain structures belonging to CATH superfamily 2.130.10.10. More are available in Section A.5 Figure 18.

(*Characteristic short-range contacts identified by row-wise feature importance*) The interval [4,18] has the highest row-wise feature importance. Figure 9 shows the contact maps of some domain structures belonging to superfamily 2.130.10.10. We see that the structures are rich in short-range contacts between amino acids with sequence separation in the range [4,18]. This is characteristic of anti-parallel $\beta$-strands present in $\beta$-propeller structures that belong to superfamily 2.130.10.10.

(*Key amino acids from repeating motifs identified by column-wise feature importance*) The amino acids $V$ and $T$ have the highest column-wise feature importance (>0.95). Amino acids $V$ and $T$ are known to be present in a repeating motif ('$YVTN$') found in many of the structures belonging to this superfamily (Chaudhuri et al., 2008). Over-

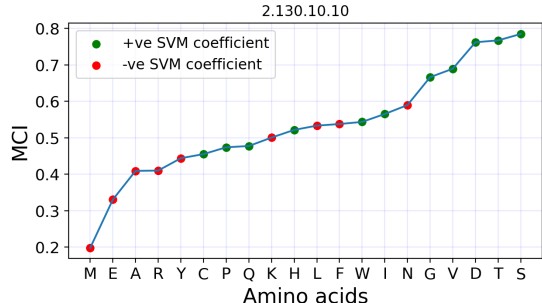

Figure 10: MCI scores of AAC feature for 2.130.10.10 vs 'others' classification.

all, the amino acids $Y, V, T$ and $N$ have greater than 0.75 MCI feature scores. Amino acid $S$ with the third highest feature importance score (>0.95) is present in a known repeating motif ('$SPDG$') found in many of

the structures belonging to this superfamily. Overall, the amino acids $S, P, D$ and $G$ have greater than 0.75 MCI feature scores.

**Using AAC features.** Figure 10 shows MCI scores of AAC for superfamily 2.130.10.10 vs 'others' classification. Amino acids $S, T$ and $D$ with the highest feature importance scores (>0.75) are present in known repeating motifs ('$SPDG$' and '$YVTN$') found in many of the structures belonging to this superfamily (Chaudhuri et al., 2008).

With AAC features, only three amino acids ($S, T$ and $D$) from the motifs ('$SPDG$' and '$YVTN$') have greater than 0.75 MCI. While with CSIC features, all the amino acids of the motif have greater than 0.75 MCI. Moreover, because CSIC uses structural information, we can identify important ranges of contact sequence separation. Thus, CSIC features offer more nuanced interpretability than the other hand-crafted or PLM-based features.

## 6 Discussion

### 6.1 Performance vs interpretability trade-off

PLM-based features outperform hand-crafted features in the CATH superfamily classification task. As ProtBERT is pre-trained exclusively using protein sequences, the high classification performance using ProtBERT-Emb suggests that there may be sequence features characteristic of a CATH superfamily. However, the specific characteristic features remain unknown due to the non-interpretable nature of the feature vectors.

On the other hand, hand-crafted features are highly interpretable, and carefully crafted features such as CSIC can achieve predictive performance comparable to some PLM-based features, such as ProtBERT-Attn. It strikes a balance between performance and interpretability and has low overfitting. Such features could be useful in inferring features that are characteristic of a CATH superfamily. The CSIC feature components are rich in information about amino acid types forming contacts and the sequence separation at which the contacts are formed. The two case studies presented in Section 5.2, illustrate that biological insights, such as characteristic features of a superfamily, can be derived using interpretable hand-crafted features like CSIC. In particular, using feature importance scores on CSIC features, we inferred characteristic features like long-range contacts in superfamily 3.40.640.10 and contacts characteristic of anti-parallel $\beta$-strands present in $\beta$-propeller structures of superfamily 2.130.10.10. Experiments on AlphaFold data confirm that CSIC features from predicted structures do not have a significant drop in performance. Also, from the error analysis of the CSIC feature, we see that misclassifications often occur between structurally similar superfamilies.

### 6.2 Future scope

Structural interpretations of characteristic features in a superfamily are possible using the CSIC features. The MCI scores for the CSIC features for three additional superfamilies are given in Section A.8. The top-ranked CSIC features of a superfamily can be further tested through directed wet-lab experiments to gain biological insights, such as the stability of the structure when these characteristic features are manipulated. However, wet-lab experiments to validate feature importance are beyond the scope of the present study.

Our hand-crafted structure-based feature engineering offers a template for other protein-related classification tasks. This may, however, need a suitable combination of domain knowledge and statistical techniques, similar to the use of contact sequence separation distribution in CSIC features.

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

# A   Appendix

## A.1   Dataset diversity

We find that the selected datasets for 45 superfamilies are very diverse. This is due to the following reasons:

- No two domain sequences, whether from the same or different superfamilies, have more than 35% sequence identity.

- Of the 45 superfamilies 6, 11, and 28 belong to 'mainly alpha' (i.e. CATH ID 1.*), 'mainly beta' (i.e. CATH ID 2.*) and 'alpha beta' (i.e. CATH ID 3.*) classes in the 1st level of CATH hierarchical classification (Figure 2). Alpha and beta denote secondary structure patterns. 'Mainly alpha' have primarily alpha helices, 'mainly beta' have primarily beta strands, and 'alpha beta' are a mixture of both.

- The dataset size for a given superfamily varies from 100 sequences to 873 sequences (Figure 2).

- The length distribution of the domain sequences varies between superfamilies (Figure 2). For example, the sequence lengths of CATH IDs 2.130.10.10 and 3.20.20.80 are $356 \pm 47$ and $348 \pm 64$ amino acids, respectively, while those of 1.10.10.60 and 2.30.30.40 are $66 \pm 18$ and $72 \pm 14$ amino acids, respectively.

- The variations of sequence lengths within some superfamilies are much higher than those for others. For example, for CATH IDs 1.25.40.10 and 3.50.50.60, the standard deviation of the sequence lengths is 103 and 98 amino acids, respectively. Meanwhile, for CATH IDs 2.30.30.40 and 2.30.42.10, the standard deviation of the sequence lengths is only 14 and 11 amino acids, respectively.

- There is no correlation between the variation of sequence lengths and the number of representative CATH domain sequences of a given superfamily.

## A.2   More details on feature engineering

### A.2.1   Hand-crafted features from sequence

From the sequence, we compute one type of feature vector that doesn't utilise any sequence order information and four other types of feature vectors that encode varying levels of sequence order information. These are discussed below.

**Amino acid composition (AAC).** As a simplistic feature, we count the number of occurrences of each of the 20 amino acid types $\mathcal{T}$, as defined in Section 2 (para 1). This results in a 20-dimensional feature vector. For a protein sequence $\mathbf{p} = (p_1, p_2, \ldots, p_L)$ of length $L$ with $p_j \in \mathcal{T}$ being one of the standard 20 amino

acids, the AAC feature $\mathbf{x}^{AAC} \in \mathbb{R}^{20}$ for $\mathbf{p}$ is computed as follows, $x_i^{AAC} = \sum_{j=1}^{L} \mathbf{1}_{\{P_j = t_i\}}, \forall i \in \{1, 2, \cdots, 20\}$. Here, $t_i \in \mathcal{T}$ is one of the defined amino acid types.

**Features that encode sequence order** We use 4 types of features that encode sequence order information, partially, into the feature vector dimensions by accounting for the relative order/position of amino acids in the protein sequence. These are *dipeptide composition (DPC), tripeptide composition (TPC), 2ordered amino acid composition (2OAAC) and 3ordered amino acid composition (3OAAC)*. DPC and TPC are existing and widely used features, while 2OAAC and 3OAAC are novel feature engineerings that are introduced in this work.

DPC is a $(20^2 =)$ 400-dimensional feature that computes the count of the contiguous 2-mers of given amino acid types in the sequence. Similarly, TPC is a $(20^3 =)$ 8000-dimensional feature that computes the count of contiguous 3-mers of given amino acid types in the sequence. In general for $k$-peptide composition ($k$PC), the count of the occurrence of a $k$-mer $(t_{i_1}, t_{i_2}, \cdots, t_{i_k})$ of amino acid types, corresponding to feature dimension $i$, in a sequence $\mathbf{p}$ is given as,

$$
\begin{aligned}
x_i^{kPC} &= x_{(i_1, i_2, \cdots, i_k)}^{kPC}, \qquad i = i_1 + \sum_{r=2}^{k} 20^r (i_r - 1) \in [20^k] \\
&= \sum_{1 \leq j \leq L-k+1} \mathbf{1}_{\{P_j = t_{i_1}, P_{j+1} = t_{i_2}, \Pi, P_{j+k-1} = t_{i_k}\}}
\end{aligned}
\tag{11}
$$

We also introduce two novel features that encode sequence order information, 2OAAC and 3OAAC.

2OAAC is similar to DPC but allows any number of residues (can be even 0) between the two amino acids, with the order of the two amino acids maintained (i.e., K_M is distinct from M_K). Consider an example sequence 'M R K P M M W A E L R V'. The ordered pair (M, R) occurs 4 times at positions $(1, 2), (1, 11), (5, 11)$, and $(6, 11)$. Meanwhile, the ordered pair (R, M) occurs twice at positions $(2, 5)$ and $(2, 6)$. Similarly, the $20^2 = 400$ dimensional 2OAAC feature vector can be computed by counting the occurrence of all $20^2$ ordered pairs of amino acids.

Likewise, 3OAAC is similar to TPC but allows any number of residues (can be even 0) between the three amino acids, with the order of the three amino acids maintained. Figure 11 illustrates how a $20^3 = 8000$ dimensional 3OAAC feature is computed.

Figure 11: For $k = 3$ in feature description in Equation (1), two occurrences of the ordered tuple (K,M,R) in a sequence of length 12. Similarly, the $20^3 = 8000$ dimensional 3OAAC feature vector for the sequence can be computed by counting the occurrence of all $20^3$ ordered 3-tuples of amino acids.

In general, for $k$OAAC, feature dimension $i$ corresponding to the ordered tuple $(t_{i_1}, t_{i_2}, \cdots, t_{i_k})$ for a sequence $\mathbf{p}$ can be computed as,

$$
\begin{aligned}
x_i^{kOAAC} &= x_{(i_1, i_2, \cdots, i_k)}^{kOAAC}, \qquad i = i_1 + \sum_{r=2}^{k} 20^r (i_r - 1) \in [20^k] \\
&= \sum_{1 \leq j_1 < j_2 < \cdots < j_k \leq L} \mathbf{1}_{\{P_j = t_{i_1}, P_{j+1} = t_{i_2}, \Pi, P_{j+k-1} = t_{i_k}\}}
\end{aligned}
\tag{12}
$$

A brute-force counting of the occurrence of $(t_{i_1}, t_{i_2}, \cdots, t_{i_k})$ in the sequence $\mathbf{p}$ will have a computational complexity of $\mathcal{O}\left(\binom{L}{k}\right)$. While using dynamic programming, it can be done in $\mathcal{O}(L)$. However, the space complexity for this feature computation is $\mathcal{O}(20^k)$.

$k$**PC loses AAC information while** $k$**OAAC retains it.** $k$PC encodes some sequence order information, but the AAC information cannot be recovered from it. This can be illustrated using a simple example. Consider the two sequences 'AARRA' and 'RRAAR'. Both the sequences have the same DPC, {'AA':1, 'AR':1, 'RR':1, 'RA':1}, while their AACs are different {'A':3, 'R':2} and {'A':2, 'R':3}. Thus, two sequences with the same $k$PC may not have the same AAC. However, if two sequences have the same $k$OAAC, then they will have the same AAC. For example, the AAC for a sequence can be recovered from its 2OAAC as follows,

$$x_{i_1}^{AAC} = \frac{1}{L-1} \sum_{i_2 \in [20]} \left( x_{(i_1,i_2)}^{2OAAC} + x_{(i_2,i_1)}^{2OAAC} \right) \tag{13}$$

In general, the $[k-1]$OAAC feature vector of a sequence can be recovered from its $k$OAAC feature vector. Thus, two sequences with same $k$OAAC will have the same $[k-1]$OAAC feature vector. However, two features with the same $k$PC may not have the same $[k-1]$PC.

### A.2.2 Hand-crafted features from structure

We propose two types of *novel feature vectors* from the 3-dimensional structure of the protein domains. For computing these features, we first compute a contact map from the protein's structure. For a protein **p** with sequence length $L$, the contact map $C$ is a square matrix of the form $C \in \{0,1\}^{L \times L}$. Where $C_{j,k} = 1$ if the distance between the centroids of the $j^{th}$ and $k^{th}$ amino acids is less than a given threshold $\theta$ in the 3D structure. We use $\theta = 7$Å (angstroms). The size of $C$ depends on the protein sequence length. We use the contact map of protein domain to compute the two types of structure-based feature vectors that are discussed below.

**Ordered contact pairs composition (OCPC).** We define OCPC as a ($20^2$ =) 400-dimensional feature that computes the count of contacts formed by given pairs of amino acid types in the protein structure. Here, the contacts are defined by the contact map. The relative order in which the two amino acids defining the contact occur in the sequence is also considered. The OCPC feature dimension $i$ for the amino acid type pair $(t_{i_1}, t_{i_2})$ from protein **p** with its contact map $C$ is computed as follows,

$$x_i^{OCPC} = \sum_{1 \le j_1 < j_2 \le L} \mathbf{1}_{\{p_{j_1}=t_{i_1}, p_{j_2}=t_{i_2}\}} \times C_{j_1,j_2}, \qquad i = i_1 + 20(i_2 - 1) \in [20^2] \tag{14}$$

**Contact separation interval composition (CSIC).** We define CSIC as $K \times 20$ dimensional feature that counts the number of contacts a given amino acid type has with any other amino acid that is within a given sequence separation range. Here, $K$ is the number of sequence separation intervals/ranges defined by the user. Let the $K$ intervals defined by the user be, $\mathcal{I} = \{[l_1, u_1], [l_2, u_2], \cdots, [l_K, u_K]\}$. The CSIC feature dimension $i$ for the amino acid type $t_{i_1}$ and interval $[l_k, u_k]$ from protein **p** with its contact map $C$ is computed as follows,

$$\begin{aligned} x_i^{CSIC} &= x_{i_1,(l_k,u_k)}^{CSIC}, \qquad i = i_1 + 20(k-1) \in [K \times 20] \\ &= \sum_{1 \le j_1 < j_2 \le L} C_{j_1,j_2} \times \mathbf{1}_{\{l_k \le j_2 - j_1 \le u_k\}} \times \mathbf{1}_{\{p_{j_1}=t_{i_1} \vee p_{j_2}=t_{i_1}\}} \end{aligned} \tag{15}$$

As in OCPC, the feature dimensions of CSIC contain two kinds of information. One is the number of contacts an amino acid type forms with other amino acids in the 3-dimensional structure of the protein. The other is how separated in the sequence are these amino acids that form contacts.

### A.2.3 Protein language model (PLM) based features from sequence

Using ProtBERT (Elnaggar et al., 2021), a pre-trained PLM, we compute two types of feature vectors from it.

**ProtBERT-Emb.** Given an input protein sequence $\mathbf{p}$ of length $L$, ProtBERT returns $L$ number of 1024-dimensional embedding vectors corresponding to each position of the input sequence. This can be viewed as a $L \times 1024$ matrix. We take the average of this matrix along the sequence length dimension to get a single 1024-dimensional embedding vector for the input sequence $\mathbf{p}$. We refer to this feature vector type as ProtBERT-Emb.

**ProtBERT-Attn.** Another feature vector that we compute from ProtBERT is using the attention-matrix from its final layer. Each layer of ProtBERT has 16 attention-heads, each generating an attention-matrix. Let's refer to attention-matrix from the final layer's $h^{th}$ attention-head as $A^h$ for the input sequence $\mathbf{p}$. $A^h$ is an $L \times L$ column stochastic matrix, i.e. $\sum_{i=1}^{l} A_{i,j}^h = 1$. We compute a 320-dimensional ($16 \times 20$) feature matrix that aggregates attention values by amino acid type. We refer to this feature vector type as ProtBERT-Attn. The ProtBERT-Attn feature dimension $i$ for amino acid type $t_{i_1}$ and attention-head $h$ is computed as follows,

$$
\begin{aligned}
x_i^{\text{ProtBERT-Attn}} &= x_{(i_1,h)}^{\text{ProtBERT-Attn}}, \qquad i = i_1 + 20(h-1) \in \left[16 \times 20\right] \\
&= \sum_{j_1=1}^{L}\left(\sum_{j_2=1}^{L} A_{j_1,j_2}^h\right) \times \mathbf{1}_{\{\mathbf{p}_i = t_{i_1}\}}
\end{aligned}
\tag{16}
$$

### A.2.4 Additional details for CSIC intervals computation

We use the expectation-maximisation (EM) algorithm for Gaussian mixture modelling (GMM) to identify the CSIC-Gaussian intervals. We discard the contacts between adjacent residues and fit $K$ Gaussian mixtures on the empirical distribution of the remaining contacts of all the structures in the train data of the OvA superfamily. For each OvA classification, we tune $K$ from 2 to 11 based on Akaike information criteria (AIC, Cavanaugh & Neath (2019)). We run the EM algorithm for at most $10^4$ steps or until the average gain in the log-likelihood lower bound is below $10^{-4}$. In our experiments, the tuned $K$ values range from 4 to 11. After identifying the $K$ intervals corresponding to the $K$ fitted Gaussians, we insert two more intervals $(0,1]$ and $[u_K + 1, \infty)$ to get the final set of intervals $\mathcal{I} = \{(0,1], [l_1, u_1], [l_2, u_2], \cdots, [l_K, u_K], [u_K + 1, \infty)\}$. An ablation study of OvA classification performance using CSIC-Gaussian with different values of $K$ is in Section A.3.1.

The computationally expensive step for CSIC-Gaussian feature calculation is the EM algorithm. Since the GMM is on a 1-dimensional distribution (please see Figure 3), one EM step has a complexity of $\mathcal{O}(n \cdot K)$ (Pinto & Engel, 2015), where $n$ is the number of samples and $K$ is the number of components for GMM. Note that the samples $n$ for the GMM are the total number of contacts from all the structures of a given CATH superfamily. This is because the GMM is fitted to the distribution of contact sequence length separations. The mean $\pm$ standard deviation of $n$ is $525 \pm 341$.

For determining the CSIC-gamma feature intervals, we use the gamma mixture modelling method from Xiong et al. (2024).

### A.3 Additional experiments

### A.3.1 Ablation study on CSIC intervals

We do an ablation study for various values of $K$ using CSIC-Gaussian features. Figure 12 shows one-vs-all classification train/test AM scores for different values of $K$ for 6 superfamilies. We do not see a consistent trend in train/test AM scores as $K$ is increased/decreased.

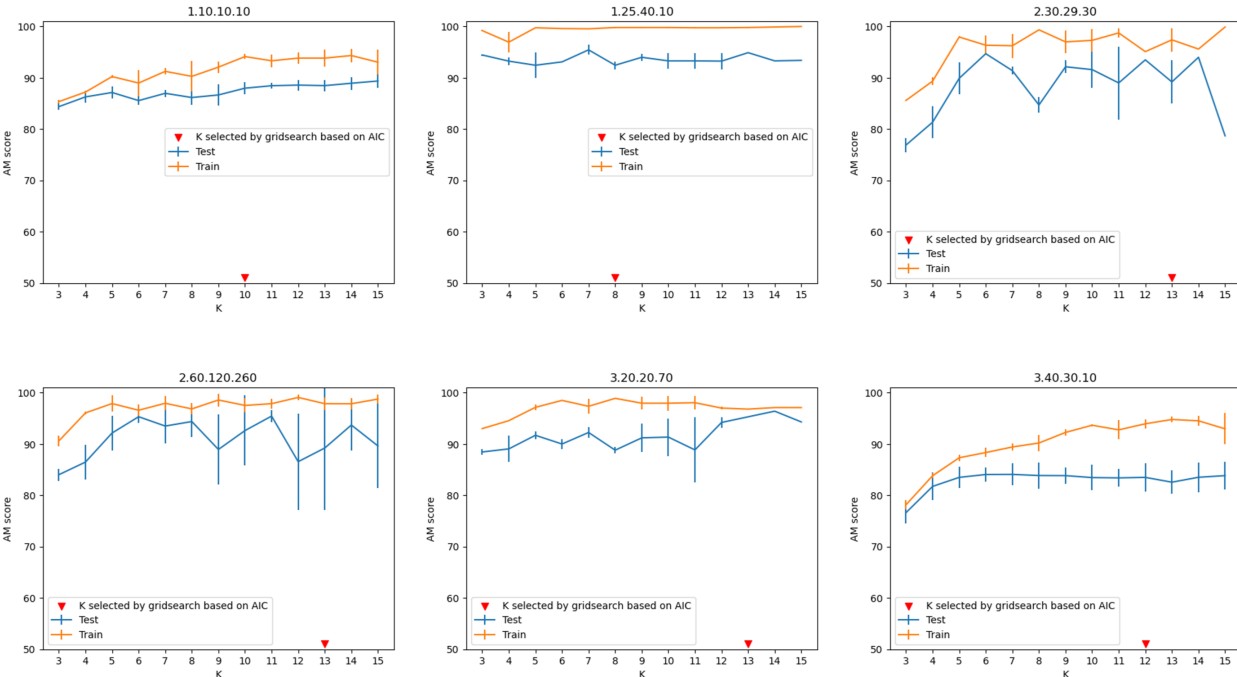

Figure 12: One-vs-all classification train/test AM scores for different values of $K$ (CSIC-Gaussian parameter) for 6 superfamilies.

### A.3.2 Statistical significance of performance differences

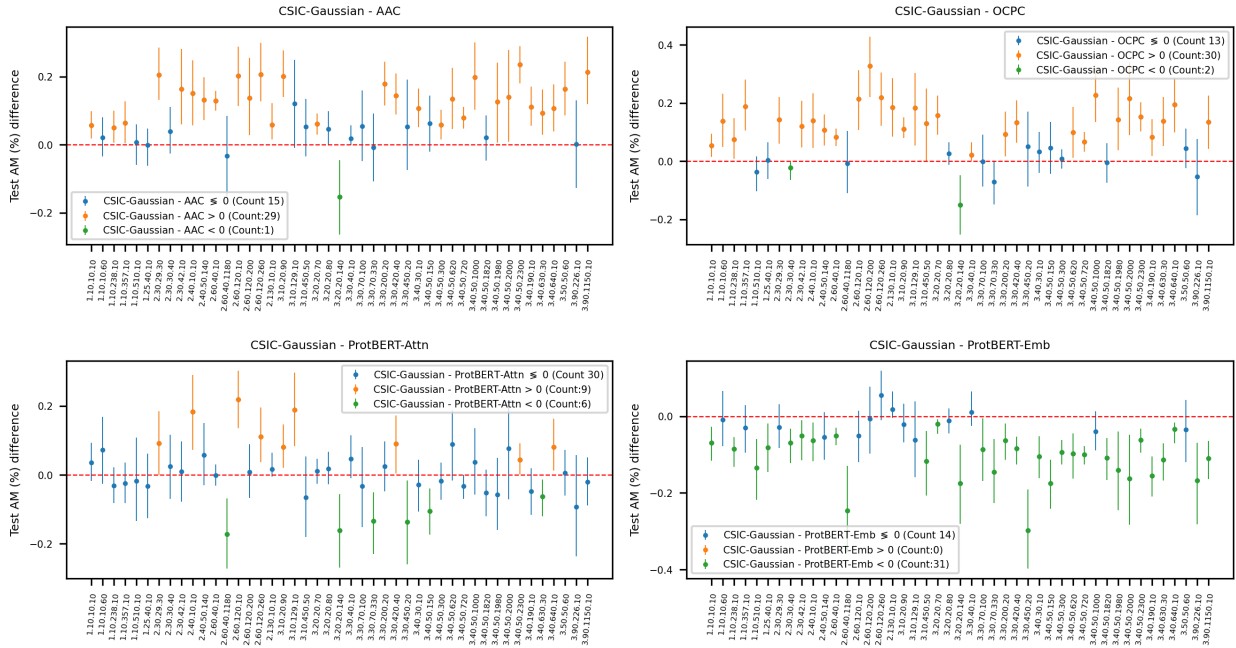

Figure 13: The error bars are the 95% confidence intervals (CI) of the test AM differences for the given feature pairs estimated using bootstrapping. $Feature1 - Feature2 > 0$ implies that the 95% CI is greater than zero. Similarly, $Feature1 - Feature2 < 0$ implies that the 95% CI is less than zero. $Feature1 - Feature2 \lessgtr 0$ implies that zero is within the 95% CI.

We performed bootstrapping on the test set to compute a 95% confidence interval for the test AM score differences of the linear SVMs trained using different feature types. We used a bootstrap sample size of 1000. Please see Figure 13. Based on these confidence intervals, we have ranked the features for each superfamily. Please see Table 3.

We observe that the ProtBERT-Emb feature consistently outperforms other features across the 45 super-families. ProtBERT-Attn and CSIC features perform comparably across the 45 superfamilies, while the rest of the hand-crafted features have relatively lower classification performances compared to them.

Thus, these experiments concur with our conclusion that CSIC features strike a balance between performance and interpretability.

### A.3.3 Applicability of CSIC to predicted structures

We created a dataset of AlphaFold predicted domain structures from the TED database (Lau et al., 2024). We collected 5350 structures in total. These proteins do not have experimentally determined structures available. The domain identifiers for these structures can be viewed here: `https://anonymous.4open.science/api/repo/cath_classification-8A0C/file/ted_alphafold_rep50.csv?v=b8158129`. These included 50 structures for each of the 45 superfamilies (total $45 \times 50 = 2250$ structures). And 3100 structures did not belong to any of the 45 superfamilies. We tested each of the binary OvA classifiers trained using CSIC features from our original dataset on the new curated AlphaFold structure dataset. See classification scores in Figure 14. The average classification AM score across the 45 superfamilies is $85.2 \pm 6.46$ (mean $\pm$ standard deviation). Thus, we do not observe a significant drop in classification performance when the CSIC feature is computed from AlphaFold structures.

Table 3: The 95% confidence intervals (CI) of test AM score differences computed using 1000 bootstrap samples. $F_1 > F_2$ implies the 95% CI of $(F_1 - F_2)$ is $> 0$. $F_1 \lesssim F_2$ implies the 95% CI of $(F_1 - F_2)$ contains 0.

**Legend:**   □: PB-Emb   ■: PB-Attn   ★: CSIC-Gamm   ☆: CSIC-Gauss
☆: OCPC   ▼: 3OAAC   ▲: 2OAAC   ▶: TPC
◀: DPC   ⊕: AAC

| CATH superfamily | Feature comparison (95% CI of test AM differences) |
|---|---|
| 1.10.10.10 | □> ■≲ ★≲ ☆> ☆≲ ▼≲ ▲> ◀≲ ⊕> ▶ |
| 1.10.10.60 | □≲ ■≲ ★≲ ☆≲ ▼≲ ▲≲ ◀≲ ⊕> ☆> ▶ |
| 1.10.238.10 | □> ■≲ ★> ☆> ☆≲ ▼≲ ▲≲ ◀≲ ⊕> ▶ |
| 1.10.357.10 | □≲ ■> ★≲ ☆> ☆≲ ▼≲ ▶≲ ▲≲ ◀≲ ⊕ |
| 1.10.510.10 | □> ■≲ ★≲ ☆≲ ☆≲ ▼≲ ▶≲ ▲≲ ◀> ⊕ |
| 1.25.40.10 | □≲ ■≲ ★≲ ☆≲ ☆≲ ◀> ⊕> ▼≲ ▶≲ ▲ |
| 2.30.29.30 | □≲ ★≲ ☆> ■≲ ☆≲ ▲> ▼≲ ▶≲ ◀≲ ⊕ |
| 2.30.30.40 | □> ■≲ ★≲ ☆≲ ☆≲ ▶≲ ▲≲ ◀≲ ⊕> ▼ |
| 2.30.42.10 | □> ■≲ ☆≲ ★≲ ▶> ☆≲ ▼≲ ▲> ◀≲ ⊕ |
| 2.40.10.10 | □> ★≲ ☆> ■≲ ☆≲ ▼≲ ▶≲ ◀≲ ▲≲ ⊕ |
| 2.40.50.140 | □≲ ★≲ ■≲ ☆> ☆≲ ▼≲ ▶≲ ▲≲ ◀≲ ⊕ |
| 2.60.40.10 | □> ■≲ ★> ☆> ☆≲ ▶≲ ▼≲ ▲≲ ◀> ⊕ |
| 2.60.40.1180 | □> ■≲ ★> ☆≲ ☆≲ ▼≲ ▲≲ ◀≲ ▶≲ ⊕ |
| 2.60.120.10 | □> ★≲ ☆> ■≲ ☆> ▼≲ ▶≲ ▲≲ ⊕> ◀ |
| 2.60.120.200 | □≲ ■≲ ☆≲ ★≲ ▶≲ ▲≲ ☆≲ ▼≲ ◀≲ ⊕ |
| 2.60.120.260 | □≲ ★≲ ☆> ■> ☆≲ ▼≲ ▶≲ ▲≲ ◀≲ ⊕ |
| 2.130.10.10 | □≲ ■≲ ★> ☆> ☆≲ ▼≲ ▶≲ ▲≲ ◀> ⊕ |
| 3.10.20.90 | □≲ ★> ☆> ■≲ ☆≲ ▶≲ ▲≲ ▼≲ ◀≲ ⊕ |
| 3.10.129.10 | □≲ ★≲ ☆≲ ■≲ ▼≲ ☆≲ ▲> ◀≲ ⊕> ▶ |
| 3.10.450.50 | □> ■≲ ★≲ ☆> ☆≲ ▼≲ ◀≲ ⊕> ▶≲ ▲ |
| 3.20.20.70 | □> ■≲ ★≲ ☆> ▼≲ ▶≲ ▲≲ ⊕> ☆> ◀ |
| 3.20.20.80 | □≲ ■≲ ☆≲ ☆≲ ★≲ ▼≲ ▶≲ ▲≲ ◀≲ ⊕ |
| 3.20.20.140 | □≲ ■≲ ★≲ ☆≲ ▼> ▶≲ ▲≲ ⊕> ☆≲ ◀ |
| 3.30.40.10 | □≲ ■≲ ☆> ★≲ ☆≲ ▼≲ ▲≲ ◀≲ ▶≲ ⊕ |
| 3.30.70.100 | □≲ ■≲ ★≲ ☆≲ ☆≲ ▲≲ ▼≲ ▶≲ ⊕> ◀ |
| 3.30.70.330 | □≲ ■> ★≲ ☆≲ ☆≲ ▶> ▼> ▲≲ ◀≲ ⊕ |
| 3.30.200.20 | □> ■≲ ★≲ ☆> ☆≲ ▶> ▼≲ ▲≲ ◀≲ ⊕ |
| 3.30.420.40 | □> ☆> ■≲ ★≲ ☆≲ ▼≲ ▲≲ ⊕> ▶≲ ◀ |
| 3.30.450.20 | □> ■≲ ★> ☆≲ ☆≲ ▶≲ ▼≲ ▲≲ ◀≲ ⊕ |
| 3.40.30.10 | □> ■≲ ★≲ ☆≲ ☆≲ ▶> ▼≲ ▲≲ ◀≲ ⊕ |
| 3.40.50.150 | □> ■> ★≲ ☆≲ ☆≲ ▼≲ ▶≲ ▲≲ ◀≲ ⊕ |
| 3.40.50.300 | □> ■≲ ★≲ ☆≲ ☆≲ ▶> ▲≲ ◀≲ ⊕> ▼ |
| 3.40.50.620 | □> ■≲ ★≲ ☆> ☆≲ ▼≲ ▶≲ ▲≲ ⊕> ◀ |
| 3.40.50.720 | □> ■> ★≲ ☆> ☆≲ ▼≲ ▲≲ ◀≲ ⊕> ▶ |
| 3.40.50.1000 | □≲ ■≲ ☆> ★≲ ☆≲ ▼≲ ▶≲ ▲≲ ◀≲ ⊕ |
| 3.40.50.1820 | □> ■≲ ★≲ ☆≲ ☆≲ ▼≲ ▶≲ ▲≲ ⊕> ◀ |
| 3.40.50.1980 | □> ■≲ ★> ☆> ☆≲ ▼≲ ▶> ▲≲ ◀≲ ⊕ |
| 3.40.50.2000 | □> ■≲ ★≲ ☆≲ ▼≲ ☆≲ ▶≲ ▲≲ ⊕> ◀ |
| 3.40.50.2300 | □> ★≲ ☆> ■> ☆≲ ▶≲ ▲> ◀≲ ⊕> ▼ |
| 3.40.190.10 | □> ■> ★≲ ☆> ☆≲ ▼≲ ▶≲ ▲≲ ◀≲ ⊕ |
| 3.40.630.30 | □> ■> ★≲ ☆≲ ▲≲ ☆≲ ⊕> ▼≲ ▶≲ ◀ |
| 3.40.640.10 | □> ★> ☆> ■≲ ☆≲ ▼≲ ▶≲ ⊕> ▲≲ ◀ |
| 3.50.50.60 | □≲ ■≲ ★≲ ☆≲ ☆≲ ▼≲ ▶≲ ▲≲ ⊕> ◀ |
| 3.90.226.10 | □> ■≲ ★> ☆≲ ☆≲ ▼≲ ▲≲ ▶≲ ⊕> ◀ |
| 3.90.1150.10 | □> ■≲ ☆> ★≲ ☆≲ ▼> ▶≲ ▲≲ ◀≲ ⊕ |

Figure 14: Mean ± standard deviation of OvA classification AM scores on CSIC-Gaussian features computed from Alphafold predicted structures. The OvA classifiers trained on the CATH dataset as described in Section 4.1 are test on predicted structures dataset. For each family 5 classifiers were originally trained from 5 random train/test splits. The mean is over the test score of each of these 5 classifiers.

### A.3.4 Mitigating variance with PCA-based dimensionality reduction

As observed in Section 5, the OvA classification performance using hand-crafted features exhibits high variance (standard deviation) across random splits. Statistically, this makes it harder to draw reliable conclusions for superfamilies with fewer samples due to the high dimensionality of the feature vectors.

To address this and test whether the variance could be controlled, we conducted experiments using Principal Component Analysis (PCA) to reduce the dimensionality of all feature vectors to 50 dimensions. The AAC feature was excluded from PCA as it is already 20-dimensional and has low variance.

We performed the OvA classification using these dimensionality-reduced feature vectors. As shown in Table 4, the classification performance is largely maintained while the standard deviation decreases significantly across all high-dimensional hand-crafted features, remaining below 3.0.

Table 4: Classification performance (Test AM scores) after applying PCA to reduce feature dimensions to 50. The average test AM scores over 5 random splits are averaged across the 45 superfamilies. Standard deviations (s.d.) are shown in parentheses. AAC is omitted as its native dimensionality (20) is already below the PCA target.

| Metric | DPC | 2OAAC | TPC | 3OAAC | OCPC | CSIC-Gauss | CSIC-Gamm | PB-Attn | PB-Emb |
|---|---|---|---|---|---|---|---|---|---|
| Test AM Avg. | 79.4 | 80.6 | 79.9 | 78.2 | 80.5 | 89.4 | 89.5 | 85.9 | 91.6 |
| (s.d.) | (2.5) | (2.7) | (2.8) | (2.8) | (2.7) | (2.1) | (2.4) | (2.4) | (2.1) |

### A.3.5 Robustness of MCI scores for CSIC-Gaussian

We have now conducted additional experiments to test the robustness of the MCI feature importance score computed for CSIC features.

We computed the CSIC-Gaussian sequence separation intervals across 5 different training data splits and the corresponding MCI scores. This was computed for 10 superfamilies with varying test scores (highest/mid/lowest). For each superfamily, we looked at the intervals with top-5 (row-wise) MCI. Please see Figure 15. We observe that the highest-ranked intervals across splits are overlapping except for superfamily 3.40.50.1980. This superfamily has a relatively very low ($<$0.5) MCI score for the top-ranked interval. The test AM score for this superfamily is also relatively low. Next, we looked at the mean ($\pm$ standard deviation) of the column-wise (amino acids) MCI scores across the data splits. Please see Figure 16. We do not see significant standard deviation in the scores, especially for features with high scores.

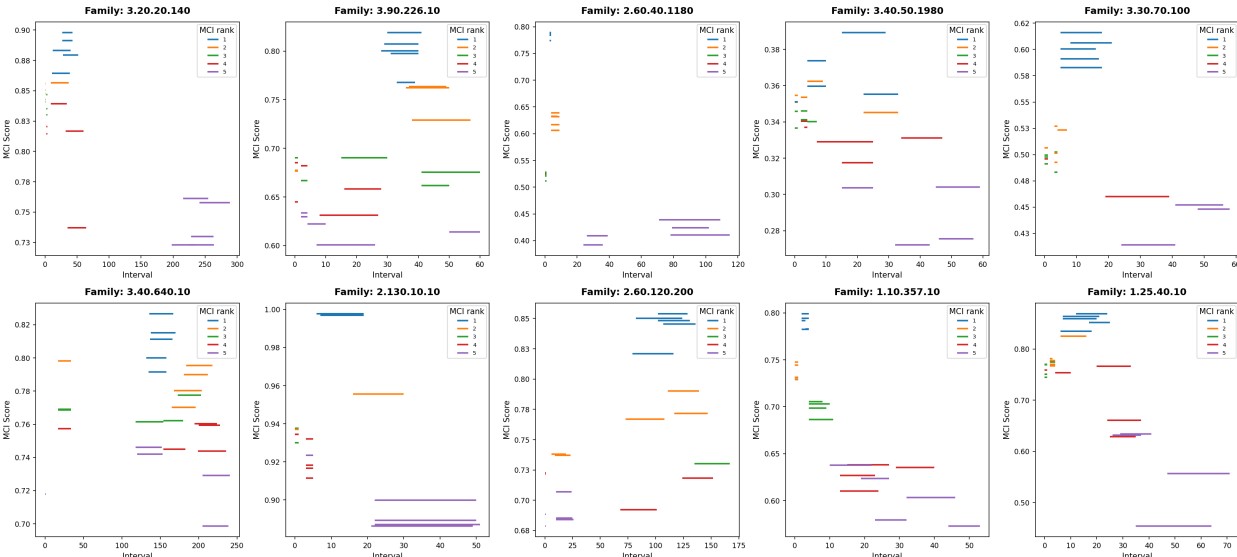

Figure 15: Each family-wise subplot shows the CSIC-Gaussian sequence-separation intervals that have the top-5 MCI scores, for the 5 different train data splits. The intervals with the same rank (via MCI) across splits are shown in the same color, and intervals with different ranks are in different colors as per the legend. The CSIC-Gaussian intervals were computed separately for each split. Ten datasets are shown here, having varied (highest/mid/lowest) test scores. We observe that either the highest-ranked intervals (in blue) across splits have significant overlap, or the top-2 intervals (blue and orange) across splits have significant overlap, except for superfamily 3.40.50.1980. This superfamily has a very low ($<$0.5) MCI score for the top-ranked interval relative to other superfamilies.

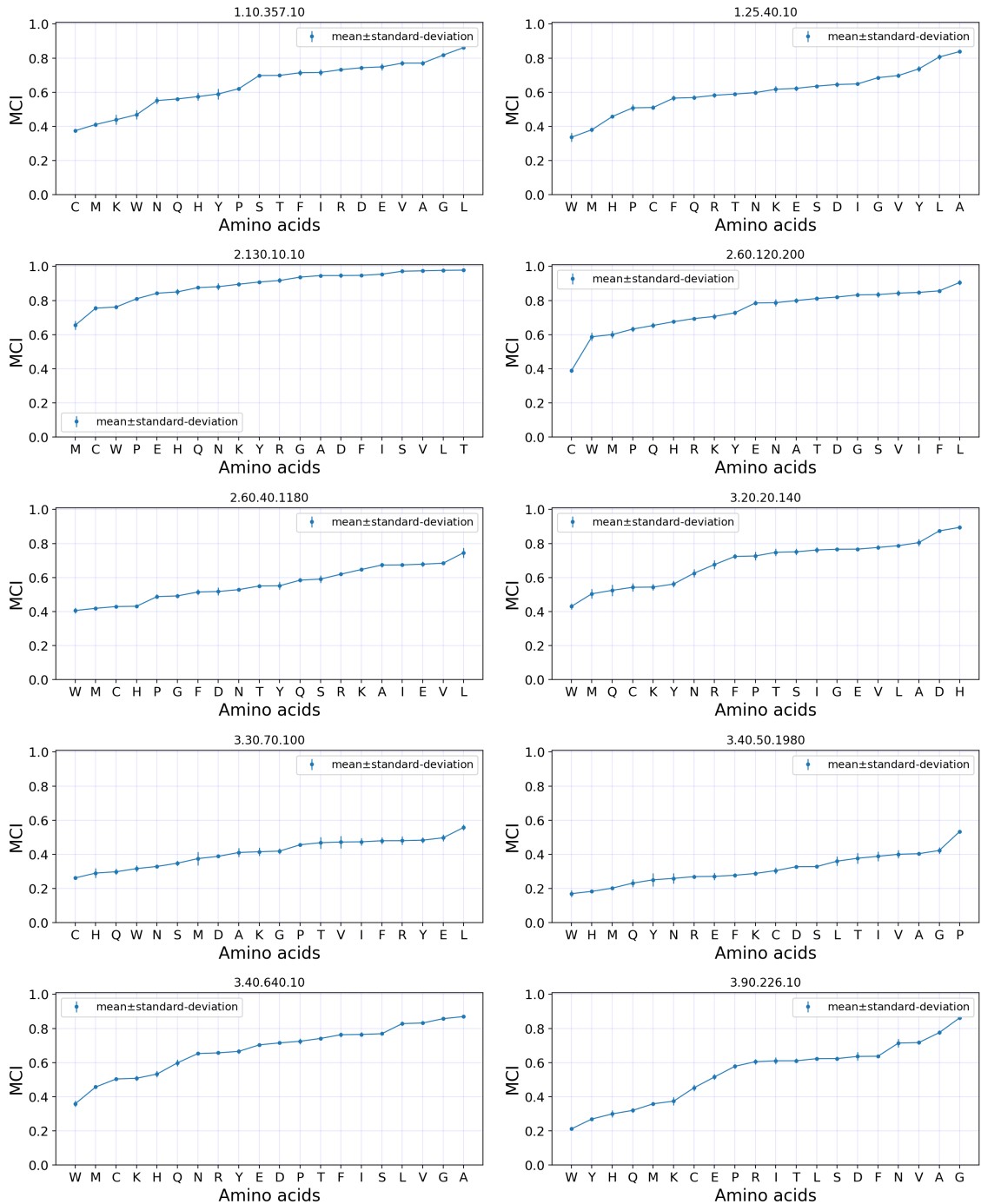

Figure 16: column-wise (amino acids) MCI of CSIC-Gaussian features for 10 superfamilies. Mean and standard deviations of the MCI scores computed over 5 different train data splits are shown.

### A.4 Hyperparameter tuning

For finding the optimal linear SVM, we implemented a dynamic, coarse-to-fine search strategy to tune the regularisation parameter $C$. First, the $C$ with the best validation AM score is searched from a broad logarithmic grid (spanning $10^{-2}$ to $10^{2}$). Next, we dynamically construct a narrower, higher-resolution parameter space localised around the best-performing $C$ to find a better value. We once again repeat this directed refinement around the current best $C$ to find a better value. Thus, we efficiently zoom in to isolate the highly optimised $C$ parameter for each specific OvA classification. Please see Figure 17, for the plots showing the train/val/test AM scores for different values of the regularisation parameter, which we tune for linear SVM.

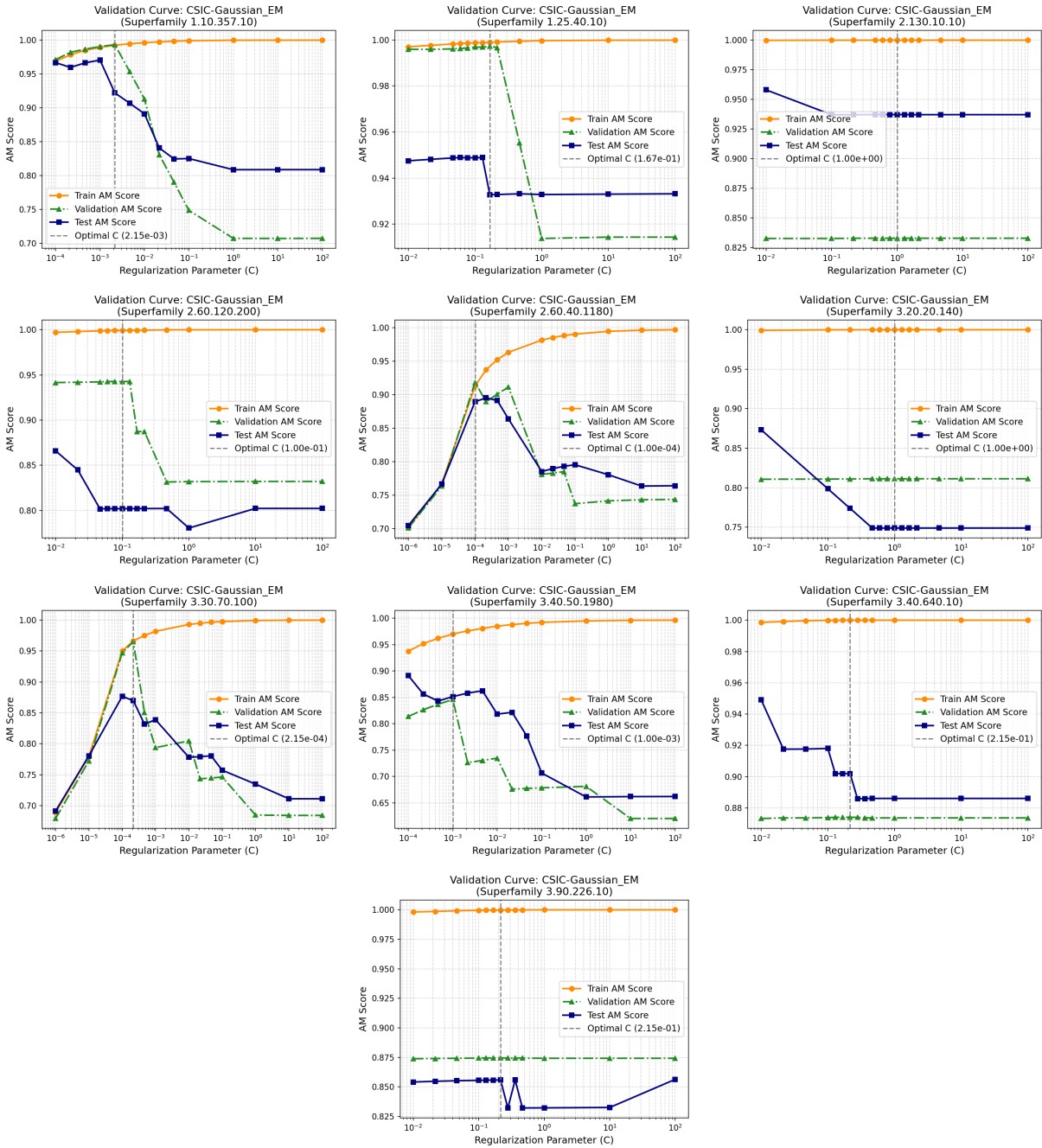

Figure 17: Train/val/test AM scores for different values of the linear SVM regularization parameter $C$ when using CSIC-Gaussian feature. Plots for 10 superfamilies are shown here.

## A.5 Contact maps

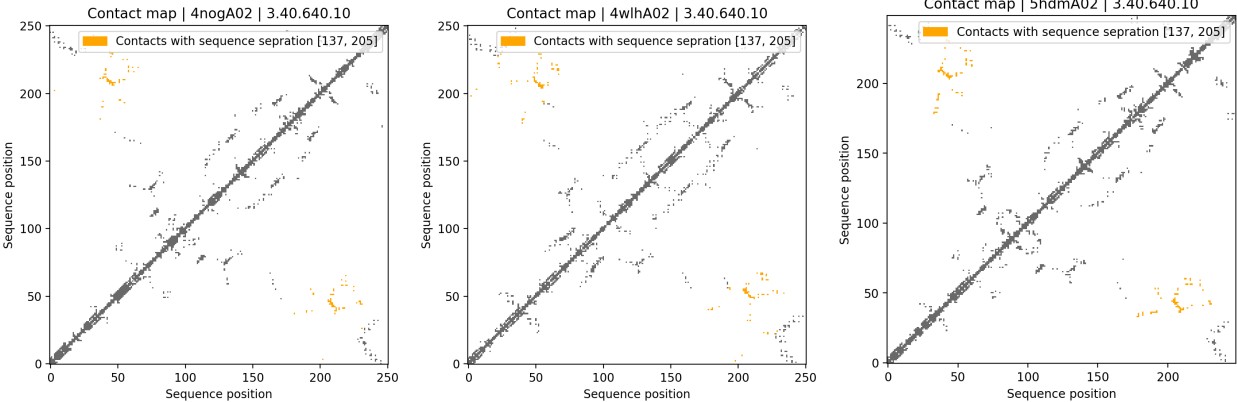

Figure 18: Contact map for 3 protein domain structures belonging to CATH superfamily 3.40.640.10

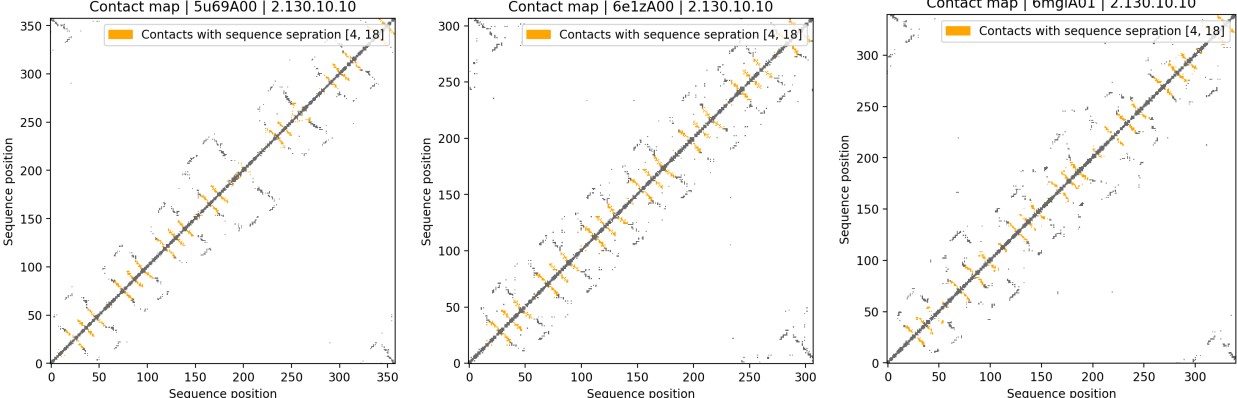

Figure 19: Contact map for 3 protein domain structures belonging to CATH superfamily 2.130.10.10

## A.6 Classification score heatmaps

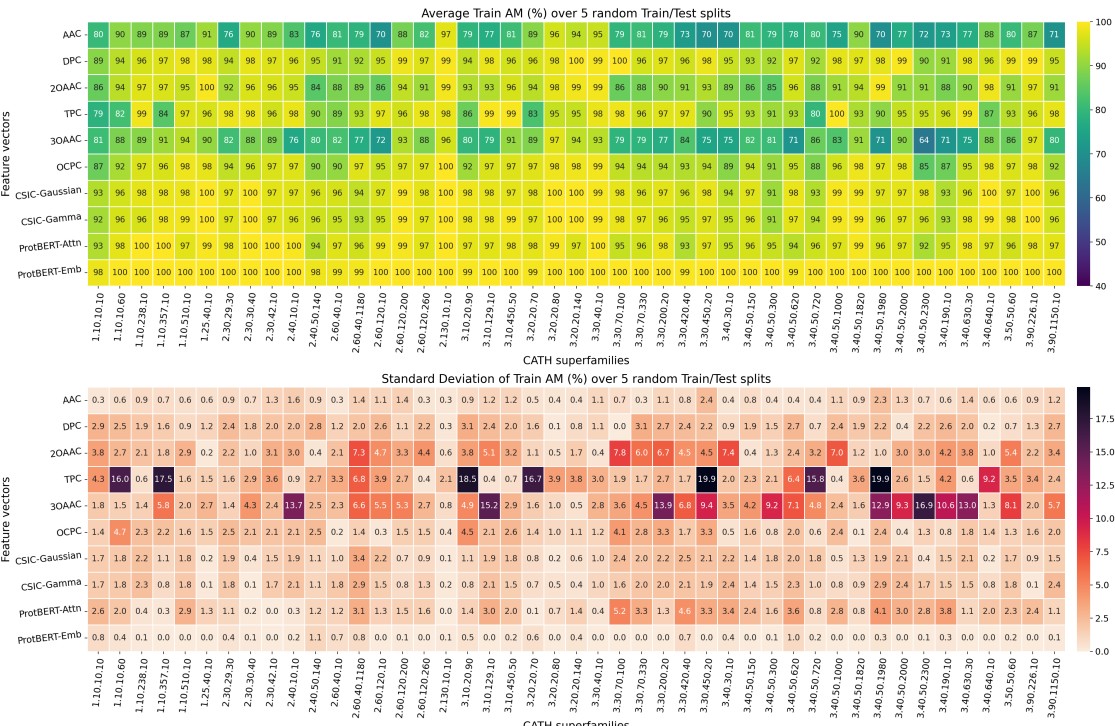

Figure 20: Training AM scores heatmap.

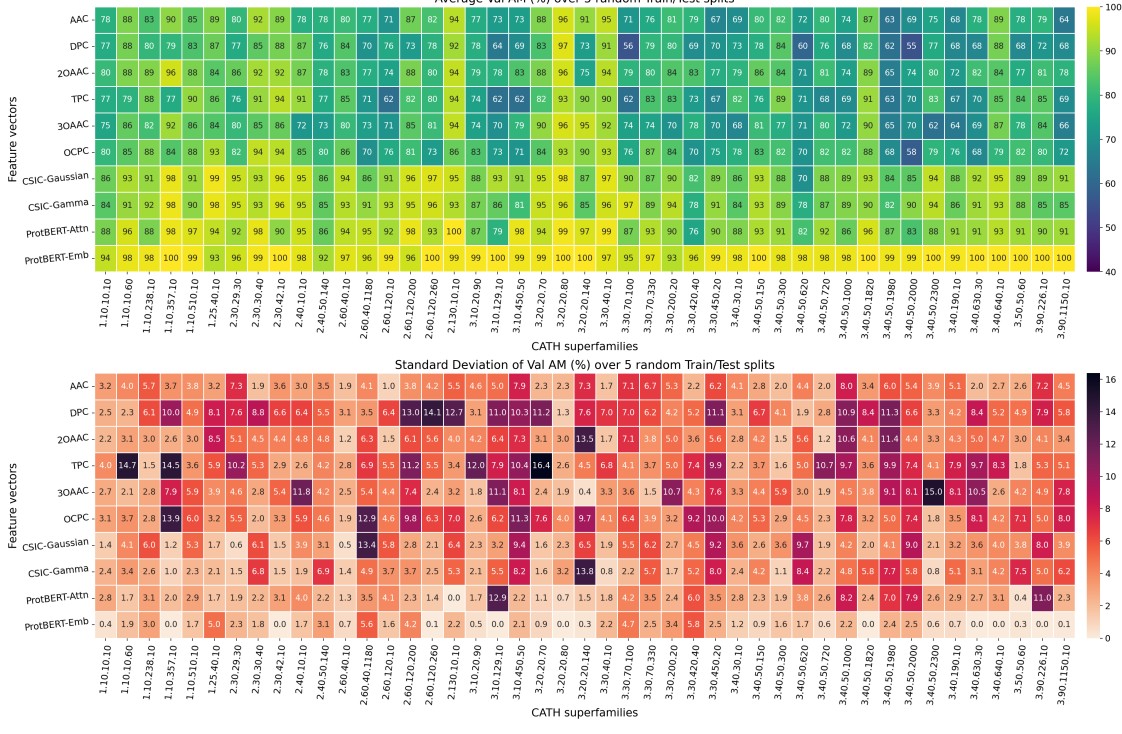

Figure 21: Validation AM scores heatmap

### A.7 Classification scores table

Table 5: Classification performance (AM scores and Accuracy) averaged across 45 superfamilies. Standard deviations (s.d.) are shown in parentheses.

| | Metric | Hand-crafted sequence-based | | | | | Hand-crafted structure-based | | | PLM-based | |
|---|---|---|---|---|---|---|---|---|---|---|---|
| | | AAC | DPC | 2OAAC | TPC | 3OAAC | OCPC | CSIC-Gauss | CSIC-Gamm | PB-Attn | PB-Emb |
| Dim. | | 20 | 400 | 400 | 8000 | 8000 | 400 | $K \times 20$ | $K \times 20$ | 320 | 1024 |
| Train | AM Avg. | 81.7 | 96.1 | 92.5 | 93.6 | 83.7 | 94.7 | 97.2 | 97.0 | 97.1 | 99.7 |
| | (s.d.) | (0.9) | (1.8) | (3.1) | (5.0) | (5.4) | (1.8) | (1.5) | (1.5) | (1.9) | (0.2) |
| | Acc Avg. | 79.5 | 92.6 | 88.9 | 87.8 | 80.2 | 91.1 | 95.0 | 94.8 | 94.9 | 99.5 |
| | (s.d.) | (1.0) | (3.5) | (4.5) | (10.0) | (10.8) | (3.5) | (2.6) | (2.7) | (3.3) | (0.4) |
| Val | AM Avg. | 79.9 | 76.3 | 82.4 | 78.7 | 78.5 | 80.7 | 90.7 | 90.1 | 91.5 | 98.1 |
| | (s.d.) | (4.2) | (6.7) | (4.6) | (6.5) | (5.2) | (5.7) | (4.2) | (4.1) | (3.2) | (1.7) |
| | Acc Avg. | 79.7 | 92.0 | 88.6 | 87.5 | 80.1 | 90.8 | 94.9 | 94.6 | 94.7 | 99.4 |
| | (s.d.) | (1.3) | (3.3) | (4.5) | (9.9) | (10.8) | (3.4) | (2.5) | (2.6) | (3.2) | (0.4) |
| Test | AM Avg. | 79.8 | 75.0 | 79.0 | 77.8 | 77.4 | 79.2 | 88.3 | 87.8 | 88.5 | 96.5 |
| | (s.d.) | (2.4) | (6.1) | (4.7) | (5.3) | (4.5) | (4.6) | (4.3) | (4.4) | (3.8) | (2.0) |
| | Acc Avg. | 79.5 | 91.9 | 88.5 | 87.4 | 80.1 | 90.6 | 94.7 | 94.5 | 94.6 | 99.4 |
| | (s.d.) | (1.1) | (3.3) | (4.4) | (9.9) | (10.7) | (3.4) | (2.5) | (2.5) | (3.2) | (0.4) |

### A.8    MCI scores of CSIC-Gaussian feature for additional superfamilies

### A.8.1    Superfamily 1.25.40.10

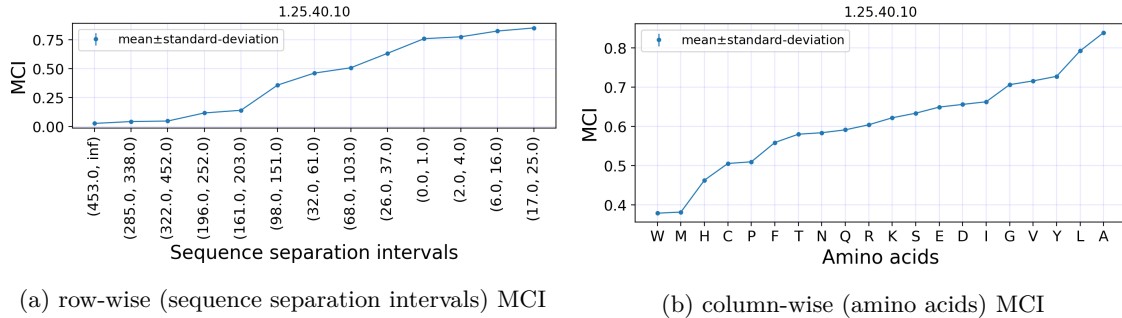

(a) row-wise (sequence separation intervals) MCI

(b) column-wise (amino acids) MCI

Figure 22: Row and column-wise MCI scores of CSIC-Gaussian $K \times 20$ feature matrix, for 1.25.40.10 vs 'others' classification.

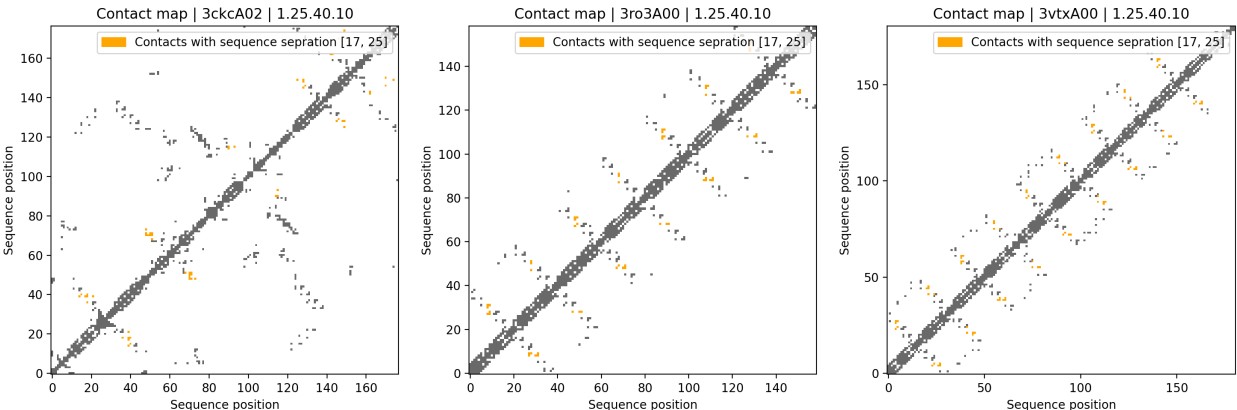

Figure 23: Contact map for 3 protein domain structures belonging to CATH superfamily 1.25.40.10.

### A.8.2    Superfamily 3.20.20.140

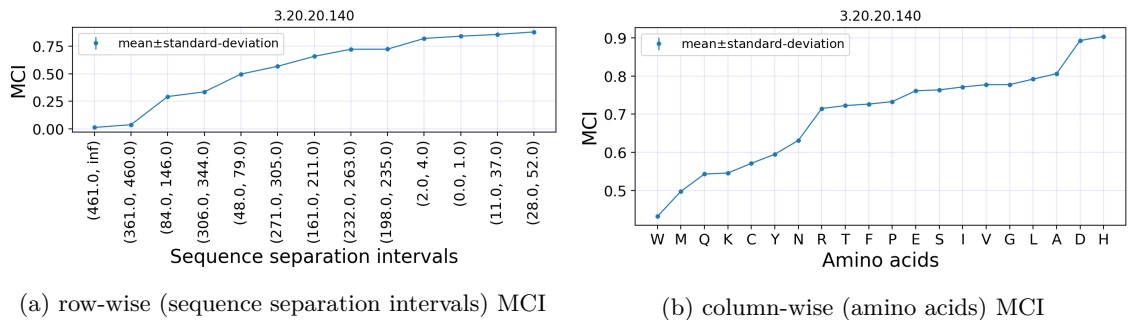

(a) row-wise (sequence separation intervals) MCI

(b) column-wise (amino acids) MCI

Figure 24: Row and column-wise MCI scores of CSIC-Gaussian $K \times 20$ feature matrix, for 3.20.20.140 vs 'others' classification.

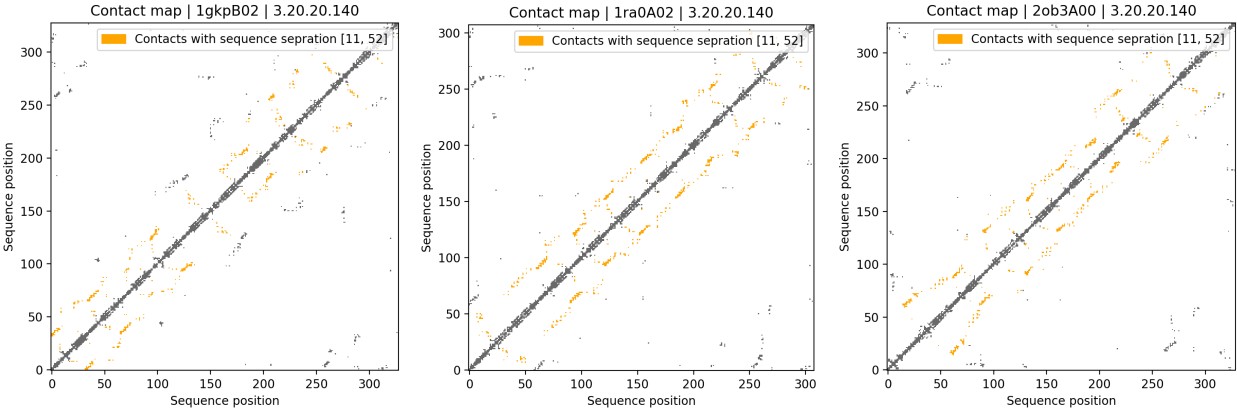

Figure 25: Contact map for 3 protein domain structures belonging to CATH superfamily 3.20.20.140.

### A.8.3 Superfamily 1.10.357.10

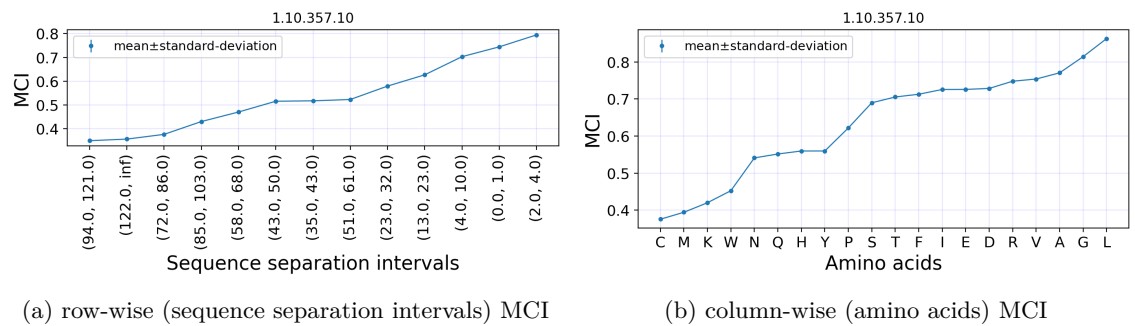

(a) row-wise (sequence separation intervals) MCI

(b) column-wise (amino acids) MCI

Figure 26: Row and column-wise MCI scores of CSIC-Gaussian $K \times 20$ feature matrix, for 1.10.357.10 vs 'others' classification.

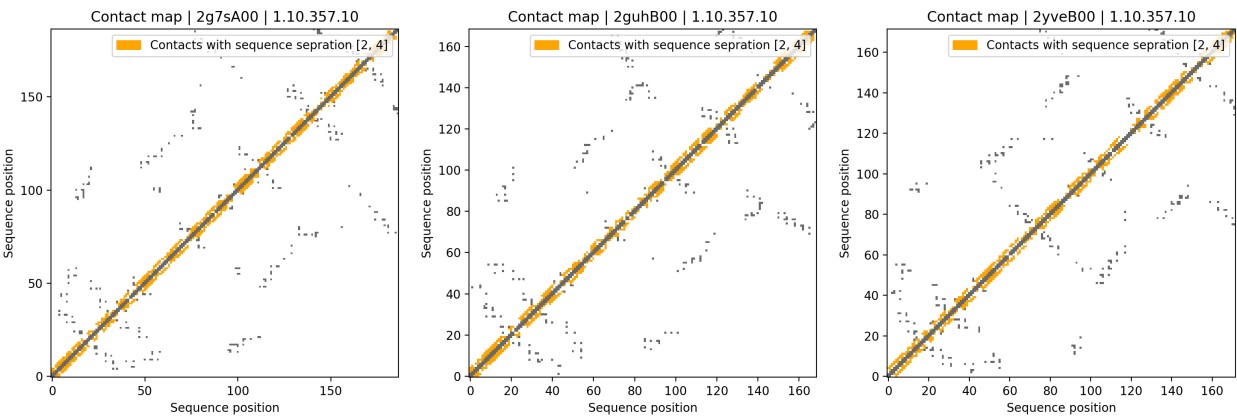

Figure 27: Contact map for 3 protein domain structures belonging to CATH superfamily 1.10.357.10.

## A.9 Error analysis plots

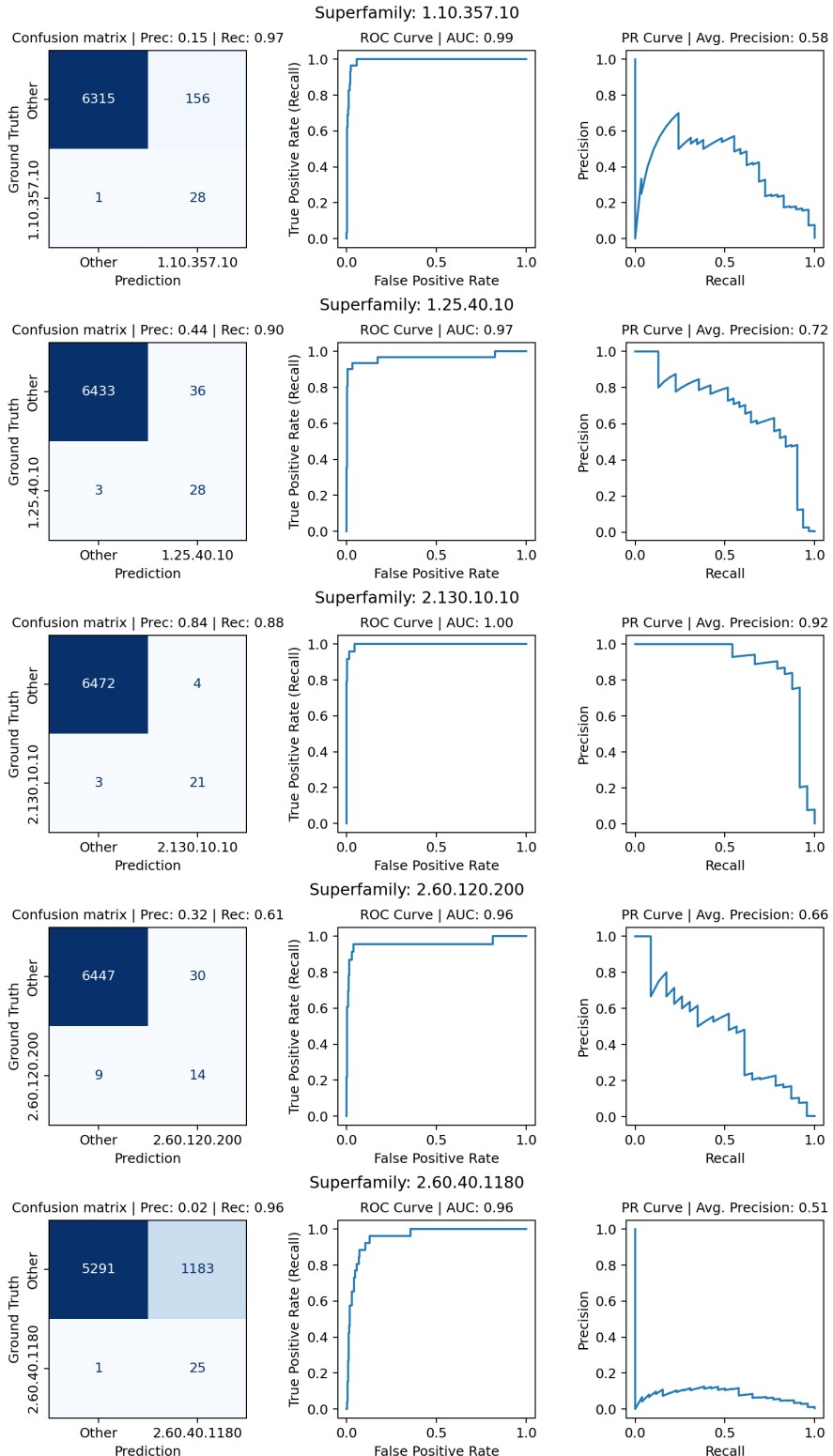

Figure 28: Test error analysis plots for CSIC-Gaussian feature. One of the train/test splits was used for computing the above. The plots for 5 more superfamilies are continued on the next page.

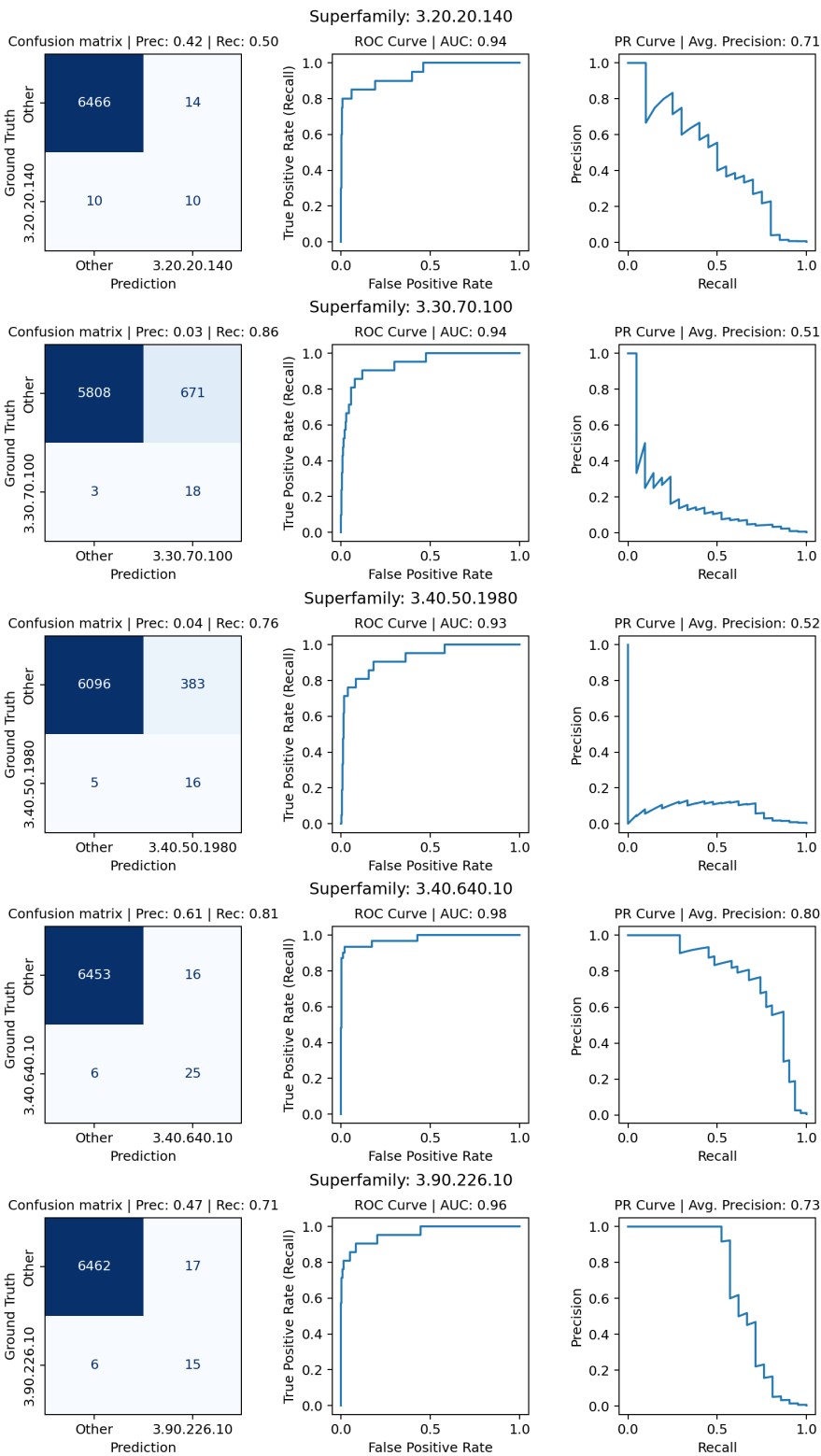

Figure 28: Figure is continued from the previous page for 5 more superfamilies. The overall 10 superfamilies shown here have varied test scores (highest/mid/lowest).

