# OpenReview forum: "Protein structural superfamily classification using hand-crafted and language model features: A performance vs interpretability trade-off"
_TMLR — Under review for TMLR_

### Review · Reviewer_Ap29 · 2026-03-07

**Summary Of Contributions:**

1. The authors propose hand-crafted (so-called interpretable) features as well as new structure- and sequence-features to predict the CATH superfamily of protein domains.
2. Their proposed features were thoroughly compared to opaque PLM-based features, demonstrating that PLM features have high predictive power but at the cost of interpretability.
3. They showed how their new hand-crafted features can be used to infer biological insights.

**Additional Comments:**

The research question is good and interesting. It is clear that the authors studied the problem thoroughly. I am just not convinced that the paper fits TMLR.

**Audience:**

No

**Audience Explanation:**

I think that the hand-crafted features can be useful, but I don't know if this is what TMLR's audience is interested in.

**Claims And Evidence:**

No

**Claims Explanation:**

I am not an expert in the field, so it's likely that I misunderstood some parts.
The authors provide evidence that CSIC is an interpretable feature representation with a reasonable predictive performance, but the authors do not provide a convincing statistical validation pipeline for biological discovery. There is no rigorous uncertainty quantification for the reported feature-importance findings. There is no multiple-testing discussion around highlighted motifs and contact intervals. There is no held-out confirmatory step. The biological "discoveries" are therefore better framed as potential interpretations of results than as statistically established findings. In any case, I cannot judge whether the findings are trivial/well-known or interesting to the community.

**Requested Changes:**

Questions
1. CSIC intervals and Figure 3: It is unclear to me how the tail should be defined. Is there any biological reasoning for this choice?
2. Changing the classification head - Gradient boosted trees. I don't understand why the authors expect gradient boosting to explain whether linear SVM overfits.
3. The handcrafted features increase the variability/variance of results. The authors should discuss the implications of this issue. For example, statistically, it would be harder to draw reliable conclusions with such noisy results (losing statistical power).
4. Considering the analysis in Figure 7 (as an example). I understand that the [4 18] interval has the highest row-wise feature importance, and this is why the authors further analyzed it. But how about the [2 2] or [0 1] intervals? They also have high feature importance scores (almost the same as the highest), but it is unclear whether they indicate anything interesting.
5. The abstract is too long; I recommend shortening it.
6. The authors should improve the statistical analysis of their results and make their experimental protocol rigorous.

---

> ### Author Response · Authors · 2026-06-05
> **Responses to reviewer Ap29 (1/2)**
>
> *We thank the reviewer for appreciating the research question of our paper.*
>
> # Responses to requested changes
>
> ## 1. CSIC Intervals and Figure 3
>
> > *`.. how the tail should be defined. Is there any biological reasoning ..?'*
>
> We would like to clarify that formally defining the tail of the distribution in Figure 3 is not crucial to our method.
> It generally refers to $P(X \ge t)$ of a 1-D random variable (please see pg. 539, DeGroot and Schervish, Prob. & Stat. 4th edition). In Figure 3, we refer to the right end of the empirical distribution of contact sequence separations within a superfamily as the tail, which has relatively lower mass than the left end. And as it appears to go to zero fast, we refer to the distribution as `light-tailed' (please see: [[link]](https://www.statisticshowto.com/heavy-tailed-distribution/)). Since short-range contacts are highly frequent, we observe a significant peak at the left end of the x-axis in the figure. While the tail of this distribution appears to taper off, the zoomed-in plot reveals multiple distinct modes that possibly correspond to conserved mid-range and long-range contacts. We leverage Gaussian mixture modeling to identify these underlying modes, thereby defining biologically meaningful, superfamily-specific sequence-separation intervals centered on the modes for CSIC feature computation. The standard deviation of the Gaussians, which are used to define the intervals, provides tolerance intervals for the sequence separations of the conserved contacts due to insertion/deletion events.
>
> Thus, *while the right end of the empirical distribution naturally forms a statistical tail, the actual biological value for the CSIC-Gaussian intervals lies in capturing the distinct modes within that region.*
>
> ## 2. Changing the classification head - Gradient boosted trees (GBTs)
>
> The experiments with gradient boosted trees were included based on feedback from a previous reviewer, who insisted on checking whether GBTs would lead to less overfitting. We agree with the reviewer Ap29’s comment that GBTs are not expected to result in less overfitting. We can redact this experiment from the paper.
>
> ## 3. Handcrafted features increase the variability/variance of results
>
> We agree with the reviewer's observation. However, we would like to point out that the standard deviations of test AM scores are particularly high for superfamilies with less number of samples, leading to greater class imbalance in OvA classification. Please see the figure: [[link]](https://anonymous.4open.science/api/repo/cath_classification-8A0C/file/test_AM_sd.png). We have now conducted experiments that show that using only a subset of important features results in a lower standard deviation of test AM. These are described below, and the paper will be revised to include this discussion.
>
> ### MCI feature subset experiment with CSIC-Gaussian:
>
> We considered only families with a standard deviation greater than 5\%, i.e. 16 families. For each of these families, we computed the row/column-wise MCI scores for the ‘CSIC-Gaussian’ features. Based on the MCI scores, we selected the top-10 amino acids and top-5 intervals, resulting in a feature subset of size $5 \times 10=50$. *Results*: We find that the standard deviations of test AM decreased across all families and remained below 5\%. For all but three families, we observed an increased test AM score. Please see the figure: [[link]](https://anonymous.4open.science/api/repo/cath_classification-8A0C/file/mci_feature_subsets.png).
> Since MCI computation is computationally intensive, we conducted experiments using PCA-based dimensionality reduction across all features.
>
> ### OvA classification with PCA-based dimension-reduced features:
>
> Using PCA, we reduced the feature dimension to 50 for all feature vectors, except AAC (which already had 20). We performed OvA classification using these dimensionality-reduced feature vectors. We observe that classification performance is maintained as the standard deviation decreases across all handcrafted features. Please see the table below:
>
> | **Metric** | **DPC** | **2OAAC** | **TPC** | **3OAAC** | **OCPC** | **CSIC-Gauss** | **CSIC-Gamma** | **ProtBERT-Attn** | **ProtBERT-Emb** |
> |:---|:---|:---|:---|:---|:---|:---|:---|:---|:---|
> | **Test AM Avg.** | 79.4 | 80.6 | 79.9 | 78.2 | 80.5 | 89.0 | 89.2 | 85.9 | 91.6 |
> | **(s.d.)** | (2.5) | (2.7) | (2.8) | (2.8) | (2.7) | (2.4) | (2.4) | (2.4) | (2.1) |
>
> Thus, we observe high variance in OvA classification performance across superfamilies with fewer samples, leading to severe class imbalance. We also find, from our experiments, that appropriate dimensionality reduction techniques can control the variance. While PCA-based dimensionality reduction is faster and relatively scalable compared to MCI, the resulting PCA features are not interpretable. On the other hand, MCI-based feature subsets are still interpretable, but are expensive to compute.

---

> ### Author Response · Authors · 2026-06-05
> **Responses to reviewer Ap29 (2/2)**
>
> ## 4. Analysis of Figure 7:
>
> > *` ... [2,2] and [0,1] intervals ... have high feature importance scores ...  it is unclear whether they indicate anything interesting.'*
>
> Indeed, the intervals [2, 2] and [0, 1] have high feature importance scores in Figure 7. The effective interval [0, 2] represents the short-range contacts with a residue that is at most 2 residues away. These are denoted by the diagonal, (i,i+1)-diagonal and (i,i+2)-diagonal entries in the contact maps (Figure 8). The number of such short-range contacts in a domain is, in general, proportional to the length of the domain sequence. The sequence lengths of domains belonging to superfamily 2.130.10.10 (which is considered in Figure 7) are, in general, higher than those of other families, as shown in Figure 2. Therefore, the domains belonging to 2.130.10.10 have a high number of contacts in [0, 2], making it a useful feature for discriminating against other families. Thus, the high importance score for the [0, 2]  interval is not surprising, but it is not as interesting a feature as the [4, 18] interval, the latter being not proportional to the sequence length.
>
> ## 5. Abstract is too long
>
> This shall be addressed in revision.
>
> ## 6. Should improve statistical analysis of the results
>
> *We respond to this in the context of the reviewer's explanation for citing a lack of evidence in the experiments.*
>
> > Reviewer comment (a):  *`There is no rigorous uncertainty quantification for the reported feature-importance findings. There is no multiple-testing discussion around highlighted motifs and contact intervals.'*
>
> We have now conducted additional experiments to test the robustness of the MCI feature importance score computed for CSIC features.
>
> We computed the CSIC-Gaussian sequence separation intervals across 5 different training data splits and the corresponding MCI scores. This was computed for 10 superfamilies with varying test scores (highest/mid/lowest). For each superfamily, we looked at the intervals with top-5 (row-wise) MCI.  Please see figure: [[link]](https://anonymous.4open.science/api/repo/cath_classification-8A0C/file/top5mci_intervals_across_splits.pdf). We observe that the highest-ranked intervals across splits are overlapping except for superfamily 3.40.50.1980. This superfamily has a relatively very low (<0.5) MCI score for the top-ranked interval. The test AM score for this superfamily is also relatively low. Next, we looked at the mean ($\pm$ standard deviation) of the column-wise (amino acids) MCI scores across the data splits. Please see figure: [[link]](https://anonymous.4open.science/api/repo/cath_classification-8A0C/file/csic-gauss_mci_aa_5plits.pdf). We do not see significant standard deviation in the scores, especially for features with high scores.
>
> > Reviewer comment (b):  *`There is no held-out confirmatory step. The biological "discoveries" are therefore better framed as potential interpretations of results than as statistically established findings.'*
>
> Statistical validation of the importance of the interpretable features is difficult, as there is no established ground truth for feature importance. It is a known limitation in computational biology that data-derived feature importance generates hypotheses for biological importance rather than definitive biological proof, which ultimately requires wet-lab experimental validation. Using case studies to validate data-driven findings is common in computational biology, for example, please see section A.2 in [[link-to-paper]](https://openreview.net/pdf?id=imcinaOHod) published recently in TMLR. The feature motifs highlighted in the two case studies in our paper are established ground truths in structural biology. Recovering these established features validates that our interpretability pipeline (CSIC + MCI) successfully captures genuine biological signals. However, finding similar supporting evidence for important features in the biological literature is difficult for many superfamilies and may not yet exist. This makes it difficult to statistically validate the pipeline's ability for biological discovery.
>
> We would also like to point out that, in our formulation of MCI, the score quantifies a feature's contribution to linear separability in one-vs-all classification. Thus, a feature with a high MCI score has a high contribution to the linear separability of classes. It is natural to expect such a feature to be significant for the respective superfamily. And if the feature is interpretable, it provides a useful clue for a domain expert to further investigate.

---

### Review · Reviewer_FJtx · 2026-04-10

**Summary Of Contributions:**

This paper studies the trade-off between predictive performance and interpretability for protein structural superfamily classification (CATH label). The experimental setting is one-vs-all classification for 45 CATH superfamilies under severe class imbalance, using linear SVMs and the arithmetic mean of sensitivity and specificity as the main metric. The paper compares a relatively broad set of baseline methods: hand-crafted sequence features (AAC, DPC, TPC, and the proposed 2OAAC/3OAAC), hand-crafted structure features (the proposed OCPC and CSIC variants), and ProtBERT-derived features (pooled embeddings and an attention-derived representation). The main empirical message is that ProtBERT-Emb performs best overall, while CSIC is the strongest interpretable hand-crafted representation and is competitive with ProtBERT-Attn. The paper also includes MCI-based case studies aimed at extracting biologically meaningful features, and an experiment using AlphaFold-predicted structures.

I found the problem well motivated and the benchmark reasonably broad within the chosen scope. I think one of the strong parts of the paper is the careful handling on interpretability rather than only accuracy, the consideration of class imbalance, and the design of contact-separation-based structure features. The main weaknesses are for its methodology and empirical evidence: the CSIC pipeline appears to use family-level information in a way that may leak held-out statistics, the evaluation is limited to 45 relatively well-populated superfamilies and an OvA protocol with many likely easy negatives (not well justified), and the interpretability claims are still supported by only a small number of case studies.

**Additional Comments:**

I found the paper thoughtful and generally well written. Table 1 is especially clear, and I appreciated that the paper tries to engage seriously with interpretability rather than treating it as an afterthought. My main concerns are concentrated around the evaluation protocol and the strength of the evidence, not the motivation or the overall direction of the work.

**Audience:**

Yes

**Audience Explanation:**

Yes. I think this paper asks a question that is clearly relevant to part of the TMLR audience: what is gained and what is lost when one replaces interpretable, biologically grounded features with high-performing but opaque learned representations in protein ML?

The comparison between PLM-based representations and hand-crafted sequence/structure features is timely, and the paper is not just another “PLMs are better” result. Its more interesting angle is that certain structure-based features, especially CSIC, may recover a nontrivial amount of predictive performance while remaining inspectable and potentially biologically useful. That is the kind of result that could interest readers working on scientific interpretability, representation learning for biology, and structure-aware machine learning.

Even where I am not fully convinced by the current evidence, I still think the topic is worthwhile and that the paper could stimulate useful discussion about how interpretability should be operationalized and validated in protein ML.

**Claims And Evidence:**

No

**Claims Explanation:**

I am close to “yes” for acceptance. The paper does support some of its more local empirical claims. In particular, it convincingly shows, on the stated 45-family benchmark, that ProtBERT-Emb outperforms the specific hand-crafted feature sets tested, and that CSIC is one of the stronger hand-crafted alternatives.

I found some confusing details and discuss below:

First, Section 4.1.2 raises a major concern about the CSIC feature construction. The paper states that, for a given superfamily, the CSIC intervals are defined by looking at the distribution of contact sequence separations across “all the structures of this family.” This means that, it suggests that the feature extractor for that superfamily uses statistics from held-out positive examples. If that is indeed how the experiments were run, then this is **a form of train/test leakage**: even if labels are not used inside the positive class, the representation itself depends on test examples. Since CSIC is one of the paper’s main contributions, this needs to be clarified and, if necessary, rerun under a strictly train-only protocol.

Second, the interpretability evidence is suggestive but still limited. The two case studies are interesting and concrete, but because interpretability is central to the paper’s framing, I would like to see more systematic validation across a broader set of superfamilies. At present, the evidence is closer to “plausible biological hypotheses can sometimes be recovered” than to a strong general claim that the proposed features reliably yield meaningful insight.

A smaller but still important concern is that many hand-crafted features are raw counts rather than normalized quantities. This makes it difficult to rule out domain length as a confounder, especially for features such as AAC and CSIC where absolute counts may partially track protein/domain size.

**Requested Changes:**

Clarify and fix the CSIC interval construction to eliminate possible train/test leakage. The interval set $I$ should be estimated using training positives only within each split, then frozen and applied to validation/test data. If the current results used all positive structures of a superfamily, the CSIC-based results should be rerun.

Strengthen the interpretability validation beyond two case studies. Because interpretability is a central contribution of the paper, I would like to see a broader quantitative or semi-quantitative evaluation across more superfamilies, for example by measuring whether top-ranked MCI features recover known motifs, contact patterns, or structural signatures. At minimum, please explain clearly how the two showcased superfamilies were selected.

---

> ### Author Response · Authors · 2026-06-05
> **Responses to reviewer FJtx**
>
> *We thank the reviewer for appreciating the motivation and direction of the work.*
>
> # Responses to requested changes
>
> > Reviewer comment (a):  *`fix CSIC interval construction to eliminate train/test leakage.'*
>
> *We acknowledge this critical flaw in our evaluation pipeline and thank the reviewer for bringing it to our attention. We have now conducted experiments to eliminate the train/test leakage.* We have used two different methods for CSIC interval construction and evaluated classification performance with both. These are,
> 1. For each train/test split, the data-driven CSIC-Gaussian/Gamma intervals were computed using only the corresponding train data. (as suggested by reviewer FJtx)
> 2. A custom user-defined CSIC interval was used of the following form,
> $$I= \\{ [0,1], [2^0+1, 2^1], ...,  [2^{i-1}+1, 2^i], ..., [2^9+1, 2^{10}], [2^{10}+1, \infty] \\}$$
>
> We see a very marginal drop in the average test AM using the data-driven interval construction method. From $88.8(\pm 4.4)$ to $88.3(\pm4.2)$ for CSIC-Gaussian, and from $88.5(\pm 3.7)$ to $87.8(\pm 4.4)$.
> Interestingly, for the second user-defined interval we see a very marginal increase in the average test AM compared to CSIC-Gaussion/Gamma, i.e. $89.4(\pm 3.8)$.
>
> The manuscript will be revised to include the above experimental results.
>
> > Reviewer comment (b):  *`how the two showcased superfamilies were selected'*
>
> The two superfamilies were selected based on their high test scores and the availability of supporting evidence from the biology literature.
>
> > Reviewer comment (c):  *`Strengthen the interpretability validation beyond two case studies … would like to see a broader quantitative or semi-quantitative evaluation across more superfamilies'*
>
> A quantitative evaluation of the importance of the interpretable features is difficult, as there is no established ground truth for feature importance. In the two case studies, we demonstrated, via contact maps, that contacts in the highest-MCI intervals are prevalent within the respective superfamily structures. Furthermore, we provided supporting evidence from the biological literature highlighting the importance of contact intervals and amino acid types with high MCI for the superfamily. However, finding similar supporting evidence for important features in the biological literature is difficult for many superfamilies and may not yet exist. Nonetheless, we will include the MCI scores for the CSIC features of additional superfamilies in the appendix, as the data provides testable hypotheses for future wet-lab work. Please see the MCI scores and contact maps for some additional superfamilies here: [[link]](https://anonymous.4open.science/api/repo/cath_classification-8A0C/file/csic_mci_extra_superfamilies.pdf).

---

### Review · Reviewer_fRhP · 2026-05-29

**Summary Of Contributions:**

Main contributions:
1. The paper provides a benchmark study of a wide variety of representation learning (PLM‑based) and hand‑crafted protein features for the challenging CATH superfamily classification task.
2. Design of the CSIC feature (contact sequence‐informed contacts) that incorporates both amino‑acid identity and sequence‑separation distribution of contacts in 3D structures.
3. Interpretability analysis of CSIC via marginal contribution feature importance (MCI) that yields biologically meaningful insights for two superfamilies (e.g., long‑range contacts in 3.40.640.10 and repeating motifs in 2.130.10.10).
4. Validation on AlphaFold predicted structures: CSIC features computed on predicted structures retain similar performance.

Key strengths:
• Extensive, systematic comparison across 45 diverse CATH superfamilies.
• Use of rigorous evaluation metrics (class‑balanced hinge loss, AM score) and statistical significance testing (bootstrapping).
• Clear demonstration that ProtBERT‑Emb dominates all other features while ProtBERT‑Attn and CSIC attain comparable performance.
• CSIC is novel in that it captures long‑range and short‑range contact patterns, which are biologically relevant.
• Demonstrates competitive predictive performance to certain PLM features while remaining interpretable.
• The case studies are compelling and illustrate that CSIC can uncover motif‑based structural patterns.	• The MCI is computed separately for rows and columns due to high dimensionality; this workaround is not fully justified.

Key Weaknesses:
• The paper is long and occasionally difficult to read; key equations and methodological details are buried in the appendix.
• Some claims (e.g., “low over‑fitting”) are not quantified in a way that makes it clear why the chosen linear SVM is optimal.
• The choice of the sequence‑separation intervals (K Gaussian components) is ad‑hoc; the paper does not provide a principled justification beyond fitting Gaussian components to the distribution.
• The MCI is computed separately for rows and columns due to high dimensionality; this workaround is not fully justified.

**Additional Comments:**

n/a

**Audience:**

Yes

**Audience Explanation:**

the paper broadly falls in the domain of interest for TMLR

**Broader Impact Concerns:**

none -- however I am not qualified to comment on potential negative outcomes of the proposed technology.

**Claims And Evidence:**

No

**Claims Explanation:**

I am unfamiliar with the standard of evidence in this domain, however:

The evidence is limited to a single metric (AM score). It would be stronger to report full ROC curves, precision–recall curves, and per‑class confusion matrices.
It could help to provide formal statistical testing (e.g., paired bootstrap) and error analysis for specific families and show training vs test curves for each feature type.

**Requested Changes:**

It would help if some of the key contributions of the paper (key equations, MCI derivation, and bootstrap methodology) were moved to the main text; keep the appendix for supplementary tables.

---

> ### Author Response · Authors · 2026-06-11
> **Responses to reviewer fRhP (1/2)**
>
> *We thank the reviewer for acknowledging the strengths of the paper.*
> # Responses to comments by the reviewer on claims/evidences
> ## Comment (a)
> >  *`It would be stronger to report full ROC curves, precision–recall curves, and per‑class confusion matrices.'*
>
> We thank the reviewer for this suggestion. We have now computed ROC curves, precision–recall (PR) curves, and per‑class confusion matrices for the OvA classification of 10 superfamilies with CSIC-Gaussian features for one of the train/test splits. Please see figure: [[link]](https://anonymous.4open.science/api/repo/cath_classification-8A0C/file/cm_roc_pr_plots.pdf). The 10 superfamilies have varied test scores (highest/mid/lowest). We find that the AUCs for the ROC curves are generally high ($\ge 0.93$) across all 10 superfamilies. However, we find that the average precision for the precision-recall curves is relatively lower for the superfamilies. Recall that our OvA classification datasets have high class imbalance (averaging 1:197). The discrepancy between ROC and PR curves is a well-studied phenomenon in such highly skewed datasets (see Daveis \& Goadrich, ICML 2006).
>
> The class-balanced hinge loss we use for training the classifiers is designed to maximise the average class-wise recall (please see Catav et al., ICML 2021), i.e. the AM score. Thus, we find high recall for the majority `other' (negative labelled) class, implying high true negatives (TNs) relative to false positives (FPs). However, due to the sheer volume of the negative class, even a small fraction of its FPs can still outnumber the (minority class) TPs, resulting in low precision.
>
> *We will include the above discussion and plots in the revision to provide a more rigorous view of the classification performance in our highly imbalanced OvA datasets.*
> ## Comment (b)
> > *`provide formal statistical testing (e.g., paired bootstrap) and error analysis for specific families and show training vs test curves for each feature type.'*
>
>  - **Statistical testing (e.g., paired bootstrap):** We have already performed bootstrapping to compute a 95\% confidence interval for test AM score differences between feature types.  Please see Section A.3.3,  Figure 12 and Table 4.
>
>  - **Error analysis:** From the ROC curves for CSIC-Gaussian features we can see that false positive rates (FPRs) are low across the OvA classifiers of the 45 superfamilies. From the per-class confusion matrices, we observe that the errors are heavily dominated by False Positives (FPs). In our OvA setup, a sample from the (negative labelled) \`other' class can belong to any of the remaining 6,630 CATH superfamilies. However, our analysis reveals that these FPs are not random misclassifications. Below are our observations from our analysis of FPs from one of the test splits:
>     - For 18 of the superfamilies, the highest number of FPs comes from the 'other' class domains that share the same *topology*  (the \`T' label in CATH, representing the same 3D structural fold. Same \`T' automatically implies same \`C' and \`A' in CATH.).  Given that there are 831 distinct topologies in the test set, the probability that the model assigns FPs to the correct topology by chance is extremely low ($< 0.0013$).
>     - We note that for 10 of the remaining 27 superfamilies, there are fewer than 5 `other' class samples available within their respective topologies, heavily restricting this specific error mode.
>     - For 7 of the remaining 17 superfamilies, the highest FPs belong to the same *architecture* (the `A' label, representing a similar spatial arrangement of secondary structures). With 38 architectures present in the test set, the probability of random assignment to the same architecture is low ($< 0.027$).
>     - *Takeaway*: This error profile demonstrates that CSIC-Gaussian feature-based linear SVM misclassifications are not random and are often between structurally similar superfamilies. Furthermore, the low FPRs demonstrate that the aggregated structure information in the hand-crafted CSIC features are sufficiently nuanced to distinguish a given CATH superfamily from many others.
>
>  - **Training vs testing curves:** We have now plotted the train/val/test AM scores for different values of the regularisation parameter $C$, which we tune for linear SVM.  Please see figure: [[link]](https://anonymous.4open.science/api/repo/cath_classification-8A0C/file/validation_curve.pdf), where we show these plots for 10 superfamilies for the CSIC-Gaussian features. We implemented a dynamic, coarse-to-fine search strategy to rigorously tune the regularisation parameter $C$. More details of this are given in the response to the next comment. Using a broader search range for $C$ may yield higher test AM scores for some superfamilies, but this would lead to longer training times, and how broad this range should be cannot be predetermined.
>
> *We will include the above error analysis and hyperparameter tuning plots in the appendix of the revised manuscript.*

---

> ### Author Response · Authors · 2026-06-11
> **Responses to reviewer fRhP (2/2)**
>
> # Responses to key weaknesses
>
> ## Comment (c):
> >  *`Some claims (e.g., ``low over‑fitting'') are not quantified in a way that makes it clear why the chosen linear SVM is optimal.'*
>
> As already discussed on page 9 (please see the 2nd item of the list below Table 2), we quantify overfitting by the difference between the train and test AM scores. Since, amongst the hand-crafted features (except AAC), CSIC shows the least train/test difference, we claim it has low overfitting. For AAC, the train/test scores themselves are relatively low.
>
> For finding the optimal linear SVM, we implemented a dynamic, coarse-to-fine search strategy to rigorously tune the regularisation parameter $C$. As already discussed in Section 4.1.1 (page 7), for each classifier, 10\% of the train set is used as a validation set for tuning the hyperparameter $C$. First, the $C$ with the best validation AM score is searched from a broad logarithmic grid (spanning $10^{-2}$ to $10^{2}$). Next, we dynamically construct a narrower, higher-resolution parameter space localised around the best-performing $C$ to find a better value. We once again repeat this directed refinement around the current best $C$ to find a better value. Thus, we efficiently zoom in to isolate the highly optimised $C$ parameter for each specific OvA classification. These additional details of the hyperparameter search will be included in the revision.
>
> ## Comment (d):
> >  *`The choice of the sequence‑separation intervals (K Gaussian components) is ad‑hoc; the paper does not provide a principled justification beyond fitting Gaussian components to the distribution.'*
>
> As discussed in the first paragraph on page 8, zooming in on the tail of the contact-sequence-separation distribution in Figure 3 reveals multiple small modes. These modes correspond to conserved mid-range and long-range contacts in the superfamily structures. We leverage Gaussian mixture modelling (GMM) to identify these underlying modes, thereby defining biologically meaningful, superfamily-specific sequence-separation intervals centered on the modes for CSIC feature computation. The standard deviation of the Gaussians, which are used to define the intervals, provides tolerance intervals for the sequence separations of the conserved contacts due to insertion/deletion events.
>
> *We will expand the discussion on page 8 to make the above motivation for using GMMs clearer to the readers.*
>
> ## Comment (e):
> > *`The MCI is computed separately for rows and columns due to high dimensionality; this workaround is not fully justified.'*
>
> As defined in equation 10 (page 22), exact feature-wise MCI computation requires evaluating the value function over $2^{|N|}$ subsets, $N$ being the set of features. While Monte Carlo sampling approximates this in linear time, the variance of the approximation bounds degrades significantly as $|N|$ grows, leading to unstable feature rankings. Thus, instead of computing MCI for each feature, we partition the feature set $N$ and compute a collective MCI score for each partition.  Recall that the CSIC features can be viewed as a $K \times 20$ matrix (see Table 1), where the $K$ rows correspond to sequence separation intervals
> and the 20 columns correspond to amino acid types. We naturally obtain interpretable row-wise and column-wise partitions of the matrix. Thus, as described in equation 11 (page 22), for computing row-wise MCI, we only require evaluating the value function over $2^K$ subsets, which is much less than $2^{|N|} = 2^{K \times 20}$. The same applies to column-wise MCI. The limitation of this approach is that we do not obtain a feature importance score for each individual feature, but rather a collective score for each set of features defined by the partitions. All features within a partition need not have equal importance.
>
> *We will expand the discussion on MCI in the revised manuscript to make our motivation for using row-wise and column-wise MCI clearer to the readers.*
>
> # Responses to requested changes
>
> > Move ... *`key equations, MCI derivation, and bootstrap methodology ... to the main text'*
>
>  We will move the key equations and details of MCI to the main text. However, we would like to keep the details of the bootstrap methodology in the appendix, as it is only a supplementary test for validating classification performances. A summary of the bootstrap experiment results is already given on page 9 of the main paper.

---

### Comment · Action_Editor_mWVW · 2026-05-29
**Author/Reviewer discussion has begun**

Dear Reviewers,

Thank you again for your work on submission 7514, "Protein structural superfamily classification using hand-crafted and language model features: A performance vs. interpretability trade-off."

All three reviews have now been posted, and the discussion phase is officially open. I'd like to ask each of you to take a moment to:

1. Read the other reviews as well as any author responses already posted.
2. Engage directly with the authors on any points needing clarification, and respond to their rebuttals where relevant.
3. Note where your assessment converges or diverges from your co-reviewers, so we can resolve any substantive disagreements before the decision stage.

The aim of this phase is for each of you to gather everything you need to feel comfortable submitting a decision recommendation. Formal recommendations can be entered starting two weeks from now, so there is good time for a thorough back-and-forth with the authors in the interim.

Please don't hesitate to reach out if anything is unclear or if I can help facilitate the discussion.

Best regards,
Action Editor, TMLR